# A TRIM21-based bioPROTAC highlights the therapeutic benefit of HuR degradation

Alice Fletcher[1] ✉, Dean Clift [2], Emma de Vries[1], Sergio Martinez Cuesta [3], Timothy Malcolm[1], Francesco Meghini[1], Raghothama Chaerkady[4], Junmin Wang[4], Abby Chiang[4], Shao Huan Samuel Weng[4], Jonathan Tart[5], Edmond Wong[1], Gerard Donohoe[1], Philip Rawlins[6], Euan Gordon[7], Jonathan D. Taylor [1], Leo James [2] & James Hunt [1] ✉

Human antigen R (HuR) is a ubiquitously expressed RNA-binding protein, which functions as an RNA regulator. Overexpression of HuR correlates with high grade tumours and poor patient prognosis, implicating it as an attractive therapeutic target. However, an effective small molecule antagonist to HuR for clinical use remains elusive. Here, a single domain antibody (VHH) that binds HuR with low nanomolar affinity was identified and shown to inhibit HuR binding to RNA. This VHH was used to engineer a TRIM21-based biological PROTAC (bioPROTAC) that could degrade endogenous HuR. Significantly, HuR degradation reverses the tumour-promoting properties of cancer cells in vivo by altering the HuR-regulated proteome, highlighting the benefit of HuR degradation and paving the way for the development of HuR-degrading therapeutics. These observations have broader implications for degrading intractable therapeutic targets, with bioPROTACs presenting a unique opportunity to explore targeted-protein degradation through a modular approach.

HuR is a ubiquitously expressed protein belonging to the RNA-binding protein family, embryonic lethal abnormal vision (ELAV), and which canonically binds mRNAs to stabilise them against degradation[1,2]. HuR overexpression correlates with inflammation, high grade tumours and poor patient prognosis due to its pleiotropic effects on tumorigenesis, which facilitate malignant transformation[3–9]. Consequently, this implicates HuR as an attractive tumour target in many cancers where HuR is overexpressed. Small molecule inhibitors of HuR have been discovered, contributing to the understanding of HuR biology while demonstrating the clinical promise of targeting HuR[5,10–14]. However, despite these efforts, the search for an effective small molecule antagonist to HuR for clinical use has proven elusive, indicating that an alternative approach is required[12–15]. The combined promise of HuR

knockdown or knockout and the advent of targeted-protein degradation as an alternative to target inhibition, implicates this as a potential strategy for modulating HuR expression[9,16–20].

This paradigm for downregulating disease-related proteins has become a reality since the emergence of proteolysis-targeting chimeras (PROTACs)—a heterobifunctional modality that simultaneously engages a specific E3 ligase and a protein-of interest for the latter's degradation via the cell's own ubiquitin system[21,22]. Small molecule (sm)PROTACs have demonstrated profound success in the degradation of a range of proteins, with an increasing number of these reaching the clinic[23]. Despite the benefits of smPROTACs, this approach is limited by E3 ligase expression within the target tissue as well as ligands which target the selected E3 ligase. A limited repertoire

[1]Biologics Engineering, R&D, AstraZeneca, Cambridge, UK. [2]MRC Laboratory of Molecular Biology, Francis Crick Avenue, Cambridge Biomedical Campus, Cambridge, UK. [3]Data Sciences and Quantitative Biology, Discovery Sciences, R&D, AstraZeneca, Cambridge, UK. [4]Centre for Genomics Research, Discovery Sciences, R&D, AstraZeneca, Gaithersburg, MD, USA. [5]Discovery Biology, Discovery Sciences, R&D, AstraZeneca, Cambridge, UK. [6]Mechanistic and Structural Biology, Discovery Sciences, R&D, AstraZeneca, Cambridge, UK. [7]Discovery Biology, Discovery Sciences, R&D, AstraZeneca, Gothenburg, Sweden. ✉e-mail: fletcher.alice@hotmail.co.uk; James.Hunt1@astrazeneca.com

of E3 ligases (VHL, CRBN and IAPs) are routinely used by smPROTACs, however, the evidence of targeted-protein degradation being dependent on the recruited E3 ligase highlights the necessity for expanding the toolbox for broader implications and enhanced selectivity[24]. Even if such limitations with E3 ligases are resolved, smPROTACs remain restricted to ligandable proteins, limiting their exploration and clinical utility.

Fortunately, alternative targeted-protein degradation approaches provide opportunities to circumvent these issues. One such technique is Trim-Away, which exploits the canonical function of E3 ligase TRIM21 to recruit antibody-bound pathogens or proteopathic agents by using antibodies against a protein of interest for subsequent ubiquitin-mediated target degradation[25–31]. As Trim-Away is reliant on exogenous antibody supply and endogenous TRIM21 expression, this technique was modified for expression of a TRIM21 Ring-B-boxed-coiled-coil (T21RBCC) fused to a single domain antibody (i.e. VHH) to form a TRIM21-based biological (bio)PROTAC[32]. By using such molecules, successful targeted degradation of GFP and GFP-tagged proteins has been demonstrated[32,33], yet a TRIM21-based bioPROTAC against an endogenous target remains to be evaluated.

BioPROTACs highlight an exciting technology that is increasingly used to control endogenous protein expression without prior modifications or a dependency on E3 ligase expression, implicating their use as tools to explore target biology, inform small molecule campaigns or as therapeutic modalities in their own right. Recognition of this has led to the discovery of bioPROTACs, and other analogous strategies such as ubiquibodies and antibody RING-mediated destruction (ARMeD), which have been successfully targeted against a multitude of proteins, including oncogenic proteins KRAS and c-Myc[34–37]. Such reports also highlight the expansion of the E3 ligase toolbox for targeted-protein degradation whereby alternative E3 ligases beyond the workhorses VHL and CRBN have been successfully implemented[34,37,38]. Owing to the relative ease of identifying target-binding domains and the rapidly expanding database of target-binding domains, this highlights a unique opportunity to explore bioPROTAC combinations against proteins-of-interest in a 'plug and play' approach.

Given the success of bioPROTACs and the difficulty in modulating HuR, this investigation was designed to evaluate whether HuR is amenable to bioPROTAC-mediated degradation. Following identification of a VHH that binds HuR and inhibits its ability to bind RNA, a TRIM21-based bioPROTAC was engineered via fusion with the T21RBCC. This HuR-targeting TRIM21-based bioPROTAC induced significant degradation of native HuR leading to anti-tumorigenic effects in a pre-clinical setting. These discoveries provide evidence of how a TRIM21-based bioPROTAC can be used to successfully target an endogenous, clinically relevant protein, highlighting the value of degradation therapeutics. Evidently, this has broad implications for degrading other intractable therapeutic targets, which can be explored rapidly through such a modular approach.

## Results

### Identification of a VHH that binds and inhibits HuR

HuR comprises three RNA-recognition motif (RRM1-3) domains important for binding target mRNAs. RRM1 is primarily responsible for mRNA binding, which is enhanced by RRM2, while RRM3 engages the polyA tail of target mRNAs[39]. Following a phage display campaign, 46 unique VHH were identified against the RRM1 and/or the RRM1 + 2 domains of HuR. The most favourable clones were selected by performing ELISA assays, expression screens and co-immunoprecipitation experiments. Three lead candidates (termed VHH$^{HuR\_8}$, VHH$^{HuR\_9}$ and VHH$^{HuR\_17}$), a weak-HuR binder (VHH$^{HuR\_18}$) and a VHH$^{Cas9}$ control (a VHH that binds Cas9 protein, acting as a negative control) were selected for subsequent studies. VHH-FLAG-HaloTag® clones were co-expressed alongside a GFP-HuR fusion, and subsequent co-immunoprecipitation confirmed both VHH$^{HuR\_8}$ and VHH$^{HuR\_17}$ engaged HuR to varying extents

(Fig. 1A). NanoBRET™ (see "Methods") – a technology analogous to fluorescence resonance energy transfer (FRET) but based on bioluminescence resonance energy transfer (BRET) – further validated these observations via identification of a HuR-specific interaction between HuR and VHH$^{HuR\_8}$ or VHH$^{HuR\_17}$, but not the negative VHH$^{Cas9}$ control (Fig. 1B). The binding kinetics describing the interaction between HuR and the lead VHH clones were measured by surface plasmon resonance (SPR). The strongest interaction with HuR was observed with VHH$^{HuR\_17}$ ($K_D$ [full-length HuR] = 0.029 μM), while VHH$^{HuR\_8}$ demonstrated a lower affinity ($K_D$ [full-length HuR] = 2.2 μM) (Fig. 1C, Supplementary Fig. 1A). The ability of each VHH$^{HuR}$ to inhibit target RNA binding by full-length HuR was next investigated via a fluorescence polarisation assay. The $K_D$ of a FITC-labelled Msi1 RNA probe with full-length HuR was first determined to be 0.0037 μM (Supplementary Fig. 1B), in agreement with published values[12,40]. By using an unlabelled RNA probe, specific HuR binding was confirmed in a competition assay with a $K_i$ of 0.004 μM. Subsequent assessment of the VHH$^{HuR}$ indicated that VHH$^{HuR\_17}$ was able to outcompete HuR-RNA binding with a $K_i$ of 0.68 μM, while the $K_i$ for VHH$^{HuR\_8}$ was higher at 7.82 μM (Fig. 1C, D). In contrast, the VHH$^{Cas9}$ control which does not bind HuR, was unable to compete with the HuR-RNA binding, $K_i > 100$ μM (Fig. 1D). Owing to the nanomolar $K_i$ for VHH$^{HuR\_17}$, competition SPR experiments using the RNA probe were undertaken. This confirmed that VHH$^{HuR\_17}$ was able to inhibit HuR by displacing bound RNA (Fig. 1E). Overall, it was evident that VHH$^{HuR\_17}$ (an HuR RRM1 binder) was the lead clone for targeting HuR owing to its target engagement, binding kinetics and inhibitory properties. However, both VHH$^{HuR\_8}$ and VHH$^{HuR\_17}$ were initially taken forward for further study.

### A TRIM21-based bioPROTAC degrades HuR

We next sought to adapt the VHH$^{HuR}$ to target endogenous HuR for degradation. According to the Trim-Away approach, a fusion between a target-binding VHH and the Fc domain from IgG can drive degradation of target proteins via recruitment of TRIM21[30]. We therefore fused VHH$^{HuR\_8}$ or VHH$^{HuR\_17}$ to an Fc domain and expressed these constructs in RPE-1 cells by mRNA electroporation. Remarkably, VHH$^{HuR\_17}$-Fc expression resulted in extensive degradation of endogenous HuR (Fig. 1F). TRIM21 was critical for HuR degradation as when a H433A mutation, which abolishes TRIM21 binding, was introduced into the Fc domain HuR degradation was mostly prevented. Consistent with its weaker binding affinity, VHH$^{HuR\_8}$-Fc expression did not affect HuR protein levels (Fig. 1F). Taken together, this highlighted that the lead clone VHH$^{HuR\_17}$ had the greatest degradation potential in a TRIM21-based bioPROTAC and was thus utilised in all future experiments. This is hereafter referred to as VHH$^{HuR}$.

The Fc fusion approach demonstrated that when HuR is in close proximity to TRIM21, it can lead to degradation of HuR. However, this strategy is dependent on sufficient endogenous TRIM21 expression[30], which may be problematic in some cell types and disease states[41–43]. We therefore reasoned that a TRIM21-based bioPROTAC, generated by genetically fusing TRIM21 directly to VHH$^{HuR}$, could also induce HuR degradation without the requirement for endogenous TRIM21. Specifically, to avoid recruitment of neo-substrates, the TRIM21 fusion was composed of only the N-terminal RBCC domain responsible for TRIM21 dimerisation and ubiquitin ligase activity, and omitting the PRY/SPRY domain[44,45] (Fig. 1G, H). It was proposed that the most effective construct would consist of an N-terminal T21RBCC, with the substrate-binding PRY/SPRY domain replaced with the VHH$^{HuR}$ (Fig. 1H). To determine the optimal orientation, the reversed order fusion was also generated.

Synthetic mRNA encoding the HA-tagged TRIM21-based bioPROTACs was transfected into three disease-relevant cell lines[9,14,19,46,47]. HuR-targeting bioPROTACs (T21RBCC-VHH$^{HuR}$ and VHH$^{HuR}$-T21RBCC) caused significant HuR degradation compared to control samples (Fig. 1I, J, Supplementary Fig. 2A–F), although it was evident that an

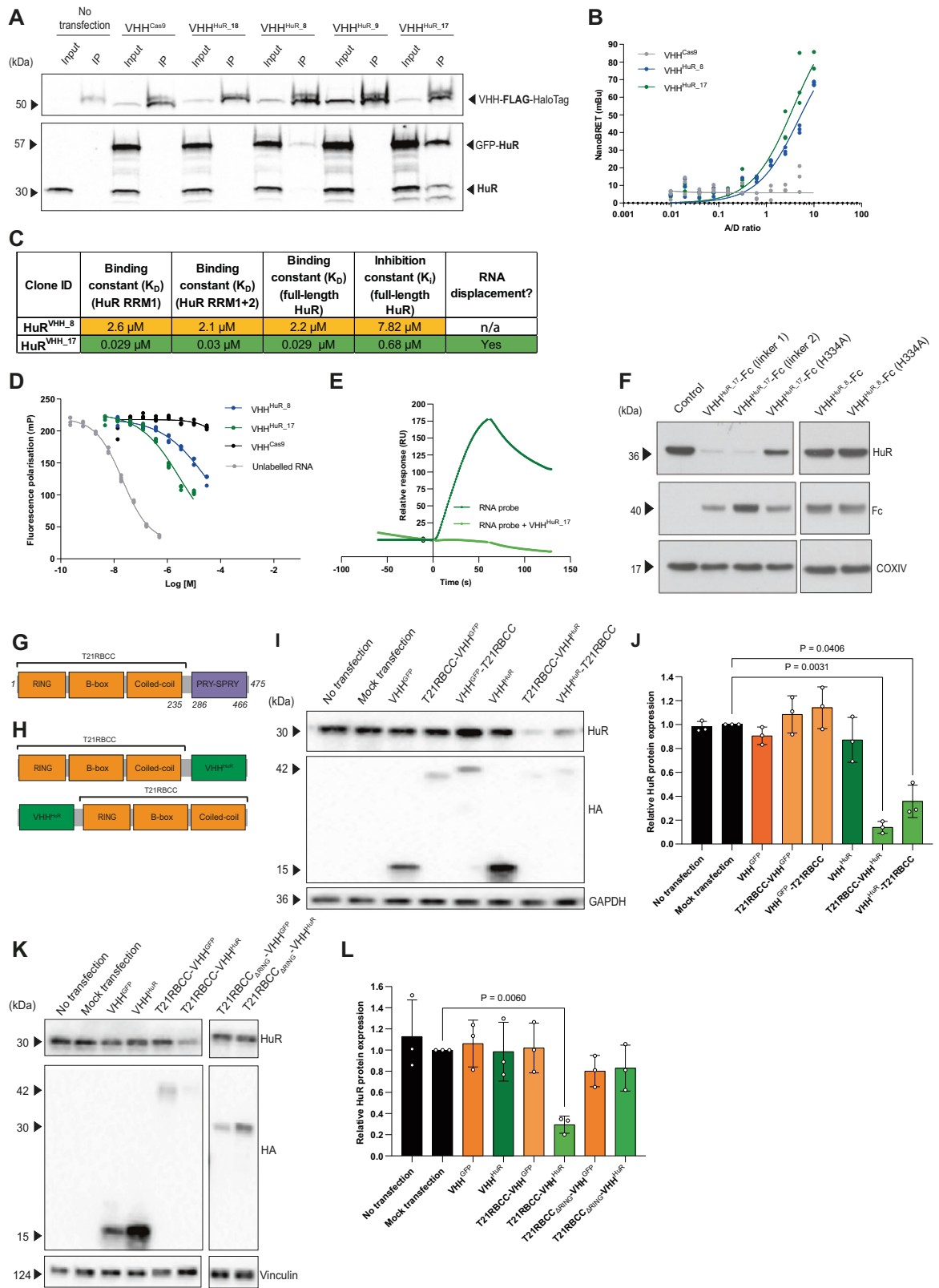

N-terminal T21RBCC (T21RBCC-VHH^HuR) degraded HuR more effi-ciently than the reverse orientation (VHH^HuR-T21RBCC), with HuR protein degradation in excess of 80% (Fig. 1I-J). In contrast, the VHH^HuR domain alone and non-HuR-targeting TRIM21-based bioPROTACs (T21RBCC-VHH^GFP and VHH^GFP-T21RBCC) did not degrade HuR (Fig. 1I, J, Supplementary Fig. 2A–F). Finally, to confirm that the observed degradation of HuR was dependent on the TRIM21 catalytic activity, a

T21RBCC lacking the active RING domain (T21RBCC_ΔRING) was gener-ated, and abolished HuR degradation (Fig. 1K, L).

For further interrogation of TRIM21-based bioPROTAC-mediated HuR degradation, stable inducible HCT116 cell lines were generated using the established ObLiGaRe doxycycline-inducible (ODIn) system[48]. The most active T21RBCC-VHH^HuR bioPROTAC was selected, together with the respective T21RBCC-VHH^GFP and VHH^HuR negative

**Fig. 1 | Identification of a VHH^HuR for use in a T21RBCC-VHH^HuR bioPROTAC to degrade HuR. A** Immunoblot of inputs and immunoprecipitation fractions from A549 cells following co-transfection of VHH-FLAG-HaloTag® (VHH^Cas9 control or VHH^HuR_8/9/18/17 clones) and emGFP-HuR. **B** NanoBRET data from HCT116 co-transfected with HuR-NanoLuc and VHH-FLAG-HaloTag® constructs (VHH^Cas9, VHH^HuR_8 and VHH^HuR_17) (n = 1, three technical replicates from distinct samples; line, non-linear fit). **C** Tabulated summary for characterisation of the two lead VHH^HuR clones – VHH^HuR_8 and VHH^HuR_17. Key: green – favourable; orange – neutral. **D** Competition assay using fluorescence polarisation to assess the ability of unlabelled RNA, VHH^HuR (VHH^HuR_8 or VHH^HuR_17) or control VHH^Cas9 (0–30 μM) to out-compete the interaction between 8 nM Msi1-FITC and 8 nM full-length HuR protein over a 5-min period. Unlabelled RNA represents complete inhibition (n = 1; four technical replicates from distinct samples; line, non-linear fit). **E** Sensorgram demonstrating the kinetic profile of 50 nM unlabelled RNA and HuR RRM1 + 2 binding in the absence/presence of 1 μM VHH^HuR_17, determined on an 8K Biacore instrument (n = 1). **F** Immunoblot following mRNA electroporation of VHH^HuR_8 or VHH^HuR_17 Fc-fusions in the RPE-1 cell line for 24 h (n = 1, representative example; linker 1 and 2 vary by a Q > E mutation in the IgG1 linker), samples derive from the same experiment and gel/blot. **G** Schematic of TRIM21 including the Ring-B-box-coiled-coil (RBCC) and canonical PRY-SPRY substrate-binding domain. **H** Depiction of the TRIM21-based, HA-tagged, bioPROTACs (T21RBCC-VHH^HuR (top) and VHH^HuR-T21RBCC (bottom)) composed of the T21RBCC and the HuR target-binding domain (VHH^HuR) in both an N- (top) and C-terminal (bottom) orientation. Representative immunoblot and densitometry following mRNA transfection of T21RBCC wildtype (**I, J**) or T21RBCC_ΔRING (**K, L**) bioPROTAC constructs in the HCT116 cell line for 18 h, samples derive from the same experiment and gel/blot. Blots shown in (**A**) are representative from n = 2 biologically independent samples per group; in F were independently repeated in HCT116 cells; in (**I**) and (**K**) are representative from n = 3 biologically independent samples per group. Data in (**J**) and (**L**) show the mean and SD from n = 3 biologically independent samples per group. Statistical significance was calculated using a one-way ANOVA and post-hoc test. Source data are provided as a Source Data file.

controls, for cell line generation. Further, the HCT116 cell line was selected owing to its demonstrable levels of HuR overexpression, and previous evidence of its use as a model in studies of HuR[14,49]. ODIn cassettes were designed for dual expression of the bioPROTAC alongside an mCherry reporter. As with transient mRNA transfections, doxycycline-induced expression of T21RBCC-VHH^HuR significantly decreased HuR while the T21RBCC-VHH^GFP and VHH^HuR had no effect on HuR levels (Fig. 2A, B). For enhanced granularity of PROTAC expression and HuR degradation, a doxycycline time-course study was completed using the T21RBCC-VHH^HuR cell line. Indirect evidence of active bioPROTAC expression (based on mCherry detection) was apparent 8 h after doxycycline induction and this continued to increase up to 48 h (Supplementary Fig. 3A–D). By 16 h post induction, a significant decrease in HuR was observed, which was sustained until at least 72 h (Supplementary Fig. 3A–D). Critically, no bioPROTAC expression or HuR degradation was observed in the absence of doxycycline (Supplementary Fig. 3A–D).

To determine if the bioPROTAC-driven degradation of HuR was occurring via the ubiquitin-proteasome system the T21RBCC-VHH^HuR cell line was co-incubated with doxycycline and proteasomal inhibitor MG132. It was observed that loss of HuR was completely inhibited by 5 μM MG132 over a 24-h period, highlighting a dependency on the ubiquitin-proteasome system for degradation (Fig. 2C, D). Indeed, MG132 was able to significantly enhance HuR levels to a similar extent in both uninduced and induced T21RBCC-VHH^HuR cells due to the inhibition of endogenous HuR turnover and thus subsequent HuR accumulation (Fig. 2C, D)[50].

### Targeted degradation of HuR decreases cell viability and colony formation

Previously, HuR has been linked to the regulation of migration, invasion, proliferation, angiogenesis and apoptosis[14,51–53]. Following demonstration of the T21RBCC-VHH^HuR bioPROTAC degrading HuR, phenotypic effects were explored using the ODIn-HCT116 cell lines to enable evaluation over an extended period of time following HuR depletion. Doxycycline-induced T21RBCC-VHH^HuR expression caused a 65% decrease in cell viability compared to the uninduced control over a 72-h period (Fig. 2E). In contrast, expression of the non-HuR-targeting bioPROTAC (T21RBCC-VHH^GFP) and VHH^HuR did not affect cell viability (Fig. 2E). Next, 3D colony formation assays were undertaken due to their mimicry of an in vivo environment. ODIn-HCT116 cell lines were embedded in agarose and incubated for 8 days to enable colony formation. Cells induced to express T21RBCC-VHH^HuR formed 77% fewer colonies than their uninduced control (Fig. 2F, G), whereas T21RBCC-VHH^GFP and VHH^HuR did not demonstrate any differences in colony formation compared to their respective uninduced control (Fig. 2F, G). In order to further investigate the mechanism of cell growth arrest on HuR degradation Western blotting of pHH3 and

pCDK2 was performed on cell lysates from experiments presented in Fig. 2A, B. A significant reduction in both these cell cycle markers was observed for doxycycline treated T21RBCC-VHH^HuR cells relative to controls (Supplementary Fig. 4A, B), consistent with a defect in proliferation as a result of cell cycle disruption[54,55].

### Global proteomics reveals effects of HuR degradation

Following observations of a T21RBCC-VHH^HuR phenotype in vitro, proteomic approaches were applied to delineate the effects of degrading HuR, in a similar manner to those previously completed for other bioPROTAC constructs[34]. Initially, HCT116 cells were transfected with mRNA encoding the T21RBCC-VHH^HuR, the VHH^HuR or the respective non-binding controls (T21RBCC-VHH^GFP or VHH^GFP). After 18 h, cells were harvested and assessed via confirmatory immunoblotting (Supplementary Fig. 5A) then subjected to LC–MS/MS analysis. A set of 6455 proteins were identified and statistically significant loss of HuR−also defined as ELAVL1 was confirmed with the active T21RBCC-VHH^HuR construct (Log$_2$FC −2.90, FDR 3.69E-26−vs. T21RBCC-VHH^GFP, Log$_2$FC −2.82, FDR 8.55E-26−vs. VHH^HuR, Log$_2$FC −2.85, FDR 6.28E-26−vs. VHH^GFP), orthogonally validating previous immunoblotting observations (Fig. 3A–C, Supplementary Data 1 and 2).

In addition to HuR, there was a subset of six proteins whose expression was significantly downregulated as a result of T21RBCC-VHH^HuR (Fig. 3A–D, Supplementary Data 3). All six of these proteins have previously been associated with HuR[2,56] representing a HuR-specific signature. From this cohort of proteins, ELAVL2/HuB, IGF2BP3 and TFAP4 were consistently decreased in all comparisons with the T21RBCC-VHH^HuR (Fig. 3A–C, Supplementary Data 1, 3). With respect to ELAV family member human antigen B (ELAVL2/HuB), it could be proposed that as the ELAVL2/HuB RRM1 and HuR RRM1 have a 74% sequence similarity ELAVL2/HuB may be directly targeted by T21RBCC-VHH^HuR. However, as ELAVL2/HuB interacts with HuR to form a complex within the cytosol of tumour cells[57,58], it is likely that ELAVL2/HuB is indirectly degraded in complex with HuR, or that they mutually stabilise each other. Previously, co-degradation in Trim-Away due to proteins forming complexes was described by Clift et al.[30]. A similar phenomenon could be proposed for IGF2BP3, which also forms a complex with HuR to co-regulate genes involved in promoting cell proliferation[59,60].

In contrast to HuR degradation by T21RBCC-VHH^HuR, the VHH^HuR – which does not degrade HuR or cause a phenotype – had limited effects on the proteome demonstrating no correlation with the altered proteome associated with T21RBCC-VHH^HuR (Fig. 3E). Another crucial question that can start to be addressed using the proteomics data is the impact of overexpression of the E3 ligase part of the bioPROTAC, T21RBCC. Overall, the proteome remained broadly unchanged by T21RBCC; however, YLPM1 depletion was confirmed to be common to

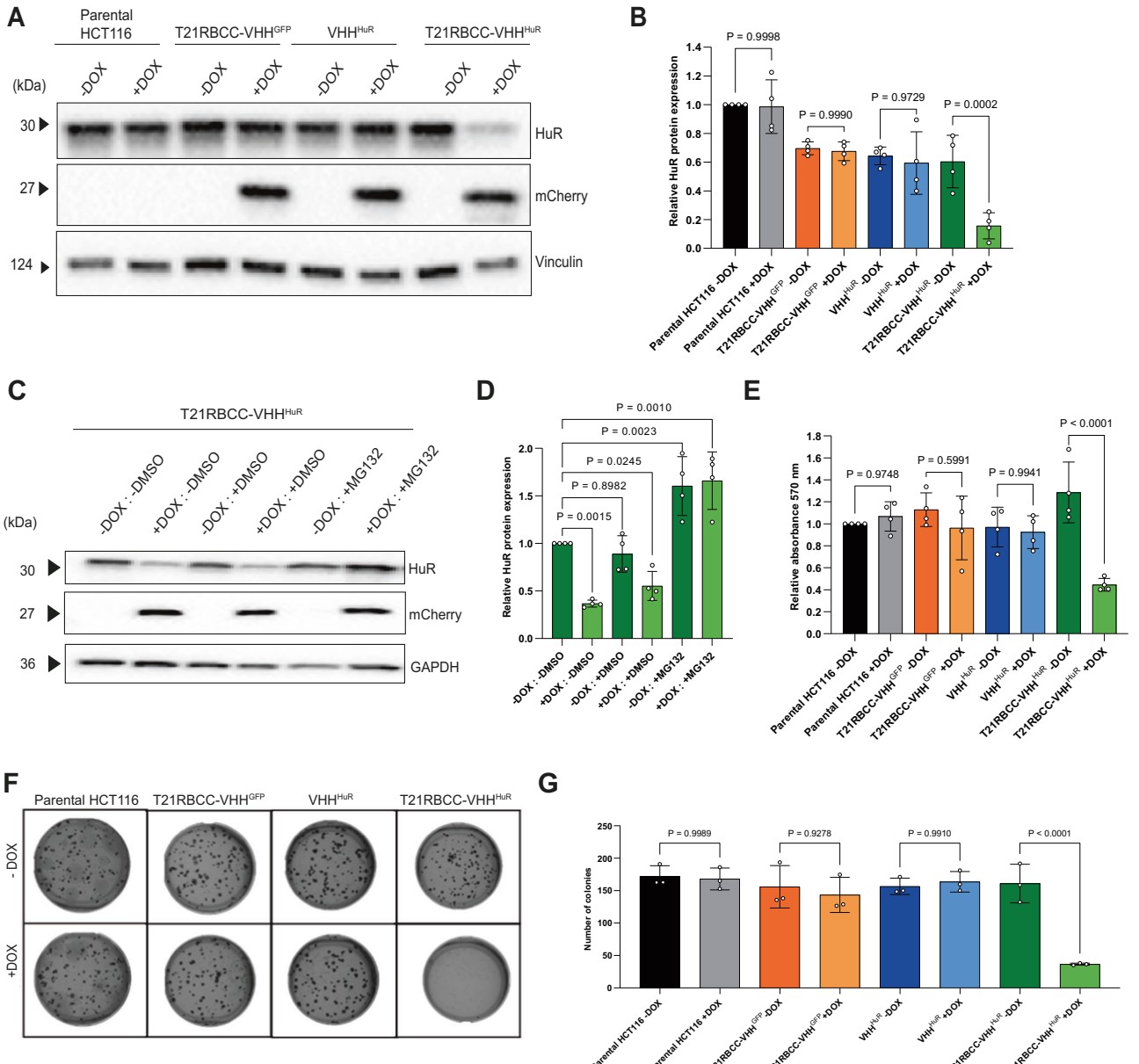

**Fig. 2 | HuR degradation causes anti-tumorigenic effects in vitro.**
**A, B** Representative immunoblot and associated densitometry of the parental HCT116 cell line and HCT116-ODIn cell lines (T21RBCC-VHH^GFP, VHH^HuR and T21RBCC-VHH^HuR) at 72 h post doxycycline. **C, D** Representative immunoblot and associated densitometry of the doxycycline-inducible T21RBCC-VHH^HuR cell line following 24 h with doxycycline and/or DMSO or 5 μM MG132. **E** Cell viability measured using the MTT assay by measuring relative absorbance (570 nm) of

parental and ODIn-HCT116 cells at 72 h post doxycycline. **F, G** Representative images and bar graph of number of colonies formed by parental and ODIn-HCT116 cells at 8 days post doxycycline in a colony formation assay. Data in **B** ($n = 3$), D ($n = 4$), **E** ($n = 4$) and **G** ($n = 3$) show the mean and standard deviation from biologically independent samples per group, as outlined. Statistical significance was calculated using a one-way ANOVA and post-hoc test. Source data are provided as a Source Data file.

---

T21RBCC (Fig. 3A–G, Supplementary Data 1, 3). Another protein modulated by T21RBCC was CCDC171, a coiled-coil domain containing protein, a notable anomaly due to its upregulation as a result of T21RBCC overexpression (Fig. 3A–G, Supplementary Data 2). However, despite these observations relating to T21RBCC-mediated effects, there was no convincing trend of change related to T21RBCC overexpression (Supplementary Data 1 and 2), in agreement with the lack of a T21RBCC-associated phenotype (Fig. 2E–G).

Having observed the direct impact of targeting HuR following 18 h transfection with the T21RBCC-VHH^HuR construct, it was important to further understand how bioPROTAC-mediated HuR degradation influences the observed T21RBCC-VHH^HuR phenotype. A kinetic proteomics study, over 72 h, was undertaken using the ODIn T21RBCC-

VHH^HuR cell line by harvesting cells at 0, 24, 48 and 72 h post induction to track global proteome changes. Due to T21RBCC-VHH^GFP and VHH^HuR lacking a phenotype, these conditions were not evaluated in this study. In agreement with confirmatory immunoblotting (Supplementary Fig. 5D), HuR degradation was observed at 24 h post induction (Log₂FC = −0.98, ns) and further decreased at subsequent time points (48 h: Log₂FC = −1.85, FDR = 1.76E-04; 72 h: Log₂FC = −1.90, FDR = 1.87 E-04) (Fig. 3H–K, Supplementary Data 4).

At 24 h post induction, there was a modestly disrupted proteome (33 proteins) compared to 48 and 72 h when the proteome was further transformed, resultant in the significantly altered expression of 171 and 172 proteins respectively (Fig. 3H–K, Supplementary Data 4 and 5). The assessment of these up- and downregulated proteins significantly

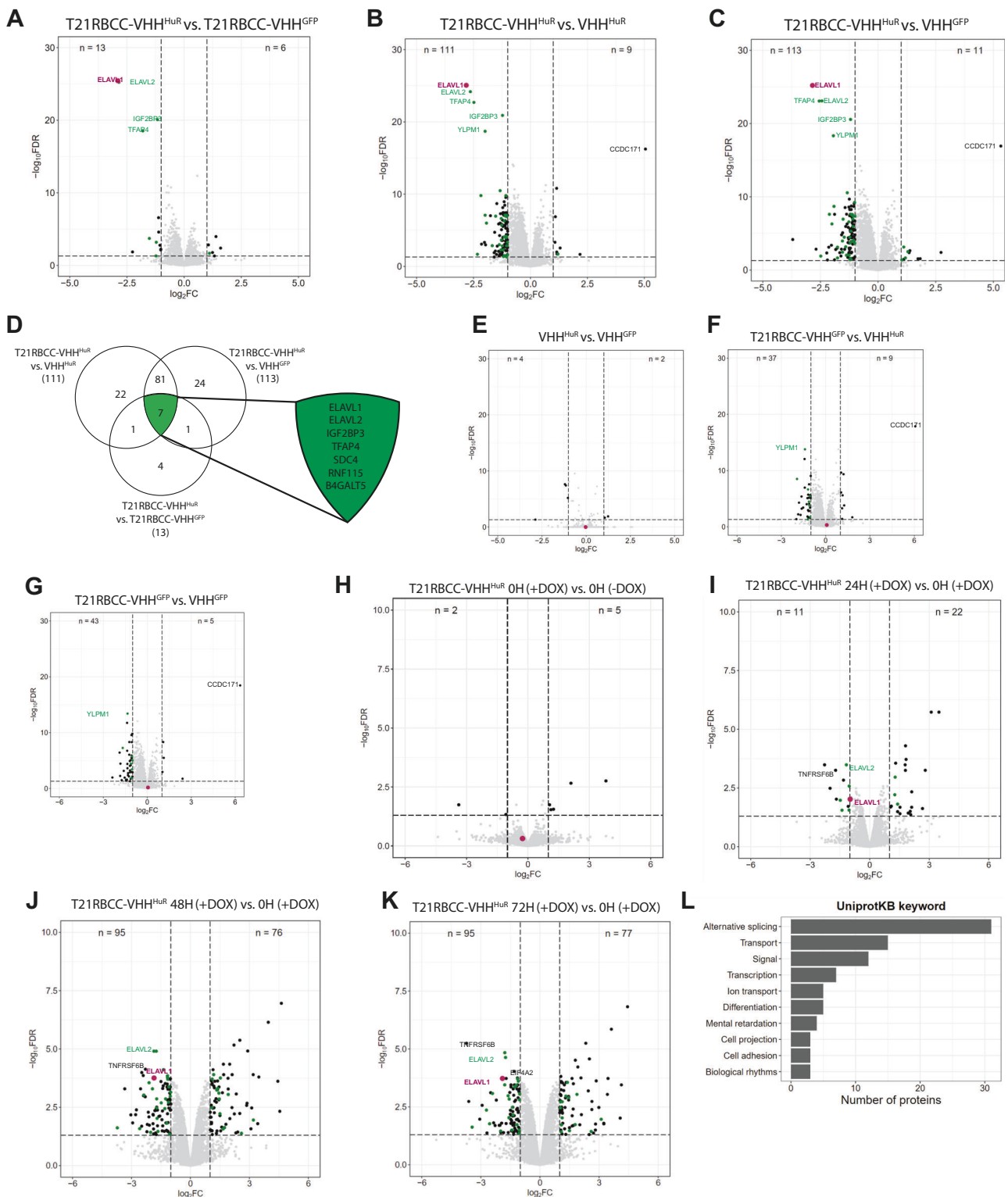

**Fig. 3 | T21RBCC-VHH$^{HuR}$ modifies the proteome.** Volcano plots displaying proteins whose expression was significantly modified by T21RBCC-VHH$^{HuR}$ in comparison with T21RBCC-VHH$^{GFP}$ (**A**), VHH$^{HuR}$ (**B**) or VHH$^{GFP}$ (**C**) upon mRNA transfections in the HCT116 cell line ($n = 3$). **D** Venn diagram of significantly downregulated proteins associated with T21RBCC-VHH$^{HuR}$ and expansion of common TRIM21·VHH$^{HuR}$-mediated proteins. Volcano plots displaying proteins whose expression was significantly modified by VHH$^{HuR}$ (**E**) or T21RBCC-VHH$^{GFP}$ (**F**, **G**) upon mRNA transfections in the HCT116 cell line ($n = 3$). Volcano plots of proteins whose expression was significantly modified in the T21RBCC-VHH$^{HuR}$ ODIn cell line

at 0 h in the presence or absence of doxycycline (**H**), and then at 24 h (**I**), 48 h (**J**) or 72 h (**K**) post doxycycline induction compared to 0 h plus doxycycline ($n = 1$). For all volcano plots, proteins were considered significantly altered with a Log$_2$FC < −1 or >1 and FDR < 0.05 (−Log$_{10}$FDR > 1.3). HuR/ELAVL1 is depicted in purple, previously described HuR-associated proteins in green and remaining proteins are in black. **L** The top 10 UniprotKB keywords linked to biological processes for the 71 significantly down- and up-regulated proteins at both 48−72 h due to T21RBCC-VHH$^{HuR}$ expression. Source data are provided as a Source Data file.

modified at both 48 and 72 h post induction highlighted changes that were consistent with a HuR-specific effect. A quarter of these proteins were previously associated with HuR (Supplementary Data 4–7). Subsequent analyses indicated the modified proteome linked to T21RBCC-VHH[HuR] was associated with biological processes such as splicing and transcription, also likely implicating the direct effect of HuR degradation on mRNA-mediated events (Fig. 3L)[1,5,6,61,62]. Additional biological processes were also highlighted with respect to T21RBCC-VHH[HuR], and included cell adhesion, differentiation and projection, transmembrane and ion transport processes - highlighting pathways which may be responsible for the T21RBCC-VHH[HuR] phenotype (Fig. 3L; Supplementary Fig. 6)[63,64].

Some notable proteins of interest whose expression was significantly decreased by T21RBCC-VHH[HuR] at 72 h post induction included EIF4A2 and TNFRSF6B, the latter of which has not previously been associated with HuR[2]. EIF4A2 is a translation initiation factor and TNFRSF6B is a decoy receptor described to protect against apoptosis[65]. Having identified IGF2BP3, TFAP4, TNFRSF6B and EIF4A2 as proteins of interest with respect to T21RBCC-VHH[HuR] in the non-kinetic and kinetic proteomic analyses, these observations were subsequently scrutinised by immunoblotting and mass spectrometry-based quantification (Supplementary Fig. 7A–K). TFAP4 expression was confirmed to be decreased in the T21RBCC-VHH[HuR] cell line however, a similar observation was made in the T21RBCC-VHH[GFP] cell line suggestive of at least a partial role of T21RBCC despite TFAP4 being associated with HuR[2]. Despite this, the lack of phenotype observed with T21RBCC-VHH[GFP] highlights that TFAP4 does not affect tumour growth. In contrast, the expression of IGF2BP3, TNFRSF6B and EIF4A2 were all confirmed to be decreased in a specific T21RBCC-VHH[HuR]-mediated response, which was enhanced over time as HuR expression became further depleted (Supplementary Fig. 7A–K).

## In vivo depletion of HuR inhibits tumour growth
Having identified significant phenotypic alterations in vitro attributed to bioPROTAC-mediated HuR degradation, mouse xenograft tumour models were established by using the T21RBCC-VHH[HuR] and VHH[HuR] ODIn cell lines within an in vivo setting. Once tumours had reached an average size of 150 mm³, half of the mice in each group were switched to a doxycycline-containing diet to induce T21RBCC-VHH[HuR] or VHH[HuR] expression (Fig. 4A). Four days after the diet change, the detection of tumoral mCherry was confirmed via in vivo imaging in mice receiving doxycycline while mCherry expression was absent in mice maintained on a standard diet (Fig. 4B, Supplementary Fig. 8A). To further validate the xenograft model, ex vivo analysis was completed on tumours resected 7 days after doxycycline initiation. As observed in vitro, the VHH[HuR] did not alter HuR expression whereas the T21RBCC-VHH[HuR] caused a significant decrease in HuR abundance (Fig. 4C, D and Supplementary Fig. 8B–E). Further, the validation of proteins identified via proteomic analyses in mouse xenograft lysates confirmed previous observations (Supplementary Fig. 7L–Q). Significantly, bioPROTAC-mediated HuR degradation rapidly arrested tumour growth with evidence of this effect already emerging at 2 days post doxycycline on day 15 (Fig. 4E and Supplementary Fig. 8F–H). Importantly, no weight loss or adverse effects were observed as a result of doxycycline treatment or T21RBCC-VHH[HuR] expression (Supplementary Fig. 8I). By the study endpoint, bioPROTAC-mediated HuR degradation had resulted in tumours that were 78% smaller than their non-induced control (Fig. 4E and Supplementary Fig. 8G, H). In contrast, the VHH[HuR]—which does not alter HuR expression—did not impact tumour growth rate (Fig. 4F and Supplementary Fig. 8F–H).

## Discussion
HuR has long been proposed as an attractive therapeutic target owing to its overexpression in cancers, which correlates with a poor patient prognosis[5–7,46]. Despite gene-targeting techniques demonstrating successful HuR knockdown or knockout, and subsequent inhibition of oncogenic activity in vitro and in vivo, such strategies are plagued by unfavourable off-target effects[9,19,20]. Furthermore, an effective small molecule antagonist against HuR for clinical use remains elusive with further investigation required[13,14]. Here, by capitalising on the success of targeted-protein degradation approaches, an HuR-binding VHH was identified and used to engineer a TRIM21-based bioPROTAC to degrade HuR. Most strikingly, HuR degradation arrested tumour growth in vivo, implicating targeted degradation of HuR as a valid alternative therapeutic approach.

While HuR degradation was evident and engineering of the TRIM21-based bioPROTAC should limit any unwanted effects, such eventualities were evaluated. It was shown that the TRIM21 RBCC is not associated with a phenotype while proteomic analyses did not highlight any major effects, setting precedence for the use of TRIM21-based bioPROTACs against a broader range of disease-associated proteins. Given that engineering of the TRIM21-based bioPROTAC demonstrated that better potency was achieved when the VHH[HuR] domain occupied the same position as TRIM21's natural target-binding PRY/SPRY domain, the exploration of TRIM21-based bioPROTACs against a broader panel of targets is feasible via a simple and rapid modular approach. Owing to the abundance of target-binding domains, the exploration of such bioPROTACs may be particularly pertinent with respect to intractable proteins for which there are currently no suitable therapeutic options.

To understand the implications of bioPROTAC-mediated HuR degradation with enhanced granularity, mass spectrometry was employed. This highlighted a HuR-specific signature was achieved with the T21RBCC-VHH[HuR] whereby only HuR and a small cohort of HuR-associated proteins were downregulated. This notion supports the use of bioPROTACs as both a research tool and therapeutic modality. Critically, over an extended period, not only was a HuR-specific signature maintained, but proteins implicated in tumorigenesis were also identified to be dysregulated. Oncogenes EIF4A2 and TNFRSF6B, which are overexpressed in tumours and described to inhibit apoptosis while promoting invasion and migration, were significantly downregulated alongside HuR and are suggestive of a pleiotropic effect[66–68]. Identification of such proteins may afford other points of intervention in cancers where HuR is overexpressed. In contrast, the VHH[HuR] was not associated with a phenotype and no meaningful effects were identified via proteomic analyses.

While this TRIM21-based bioPROTAC elicits specific HuR degradation with limited TRIM21-mediated effects, the most striking observation is the rapid and sustained tumour growth arrest in vivo. These findings highlight the potential of HuR degradation as an alternative to HuR inhibition, likely owing to the implications of sustained HuR overexpression even in the presence of an inhibitor, a concept previously observed for other targets[16,18,69]. It could be postulated that due to the observed high affinity of RNA with HuR, the occupancy-driven state of inhibition is particularly challenging as HuR inhibitors are easily outcompeted by canonical mRNA binding, which can only be overcome by high levels of inhibitor[70]. Meanwhile, HuR degradation occurs via an event-driven strategy whereby transient targeting of HuR by the bioPROTAC enables sufficient levels of target depletion, which can be maintained until de novo synthesis[38,70]. Future studies could look to extend observations beyond the arrest of proliferation of the primary tumour, to look at the ability of these bioPROTACs to also disrupt the process of metastasis[71]. Perhaps through using targeting approaches that have started to be explored in other systems[72]. Collectively, this study reveals engineering of a T21RBCC-VHH[HuR] bioPROTAC which can degrade endogenous HuR for a profound arrest of tumour growth. This study sets precedence for developing HuR-degrading therapeutics—either by re-purposing HuR small molecule inhibitors or for the delivery of bioPROTACs as therapeutics once technologies allow[73,74]. More broadly, this study also

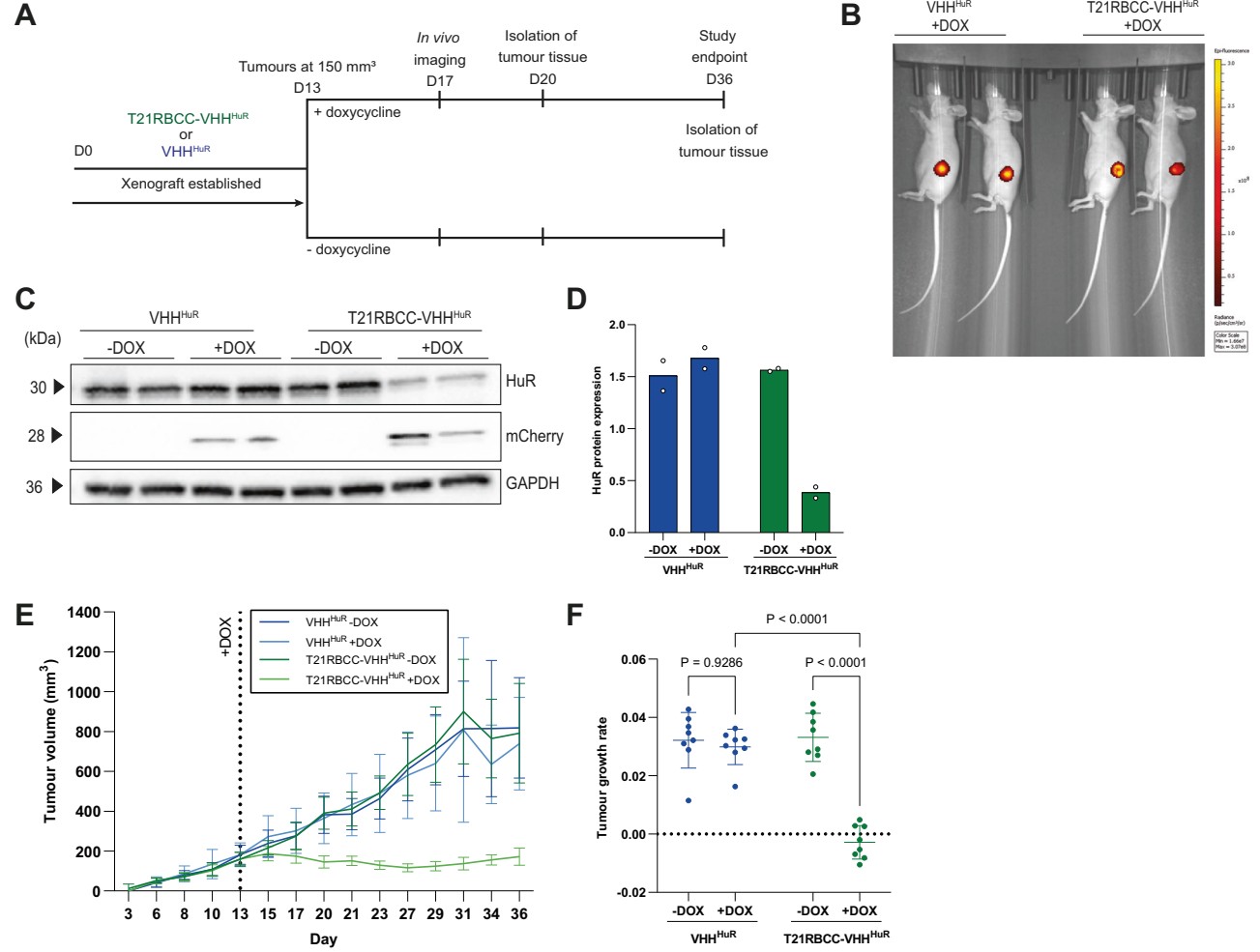

**Fig. 4 | T21RBCC-VHH^HuR arrests tumour growth in vivo. A** Schematic of experimental design for evaluating doxycycline-inducible T21RBCC-VHH^HuR and VHH^HuR xenograft models. HCT116-ODIn T21RBCC-VHH^HuR and VHH^HuR cell lines were engrafted in mice at day 0 and left to establish until day 13 (average tumour size 150 mm³) when tumour-bearing mice were size-matched and randomly assigned into experimental groups and switched to a doxycycline diet. On day 17, mice on the doxycycline diet underwent in vivo imaging for mCherry expression and at day 20, two mice from each group were culled for isolation of tumour tissue. **B** In vivo imaging of tumoral mCherry in mice implanted with either the VHH^HuR or T21RBCC-VHH^HuR cell line at 4 days post doxycycline induction. **C, D** Representative immunoblot and associated densitometry of VHH^HuR or T21RBCC-VHH^HuR xenograft lysates (−/+ doxycycline). Graphs of tumour volumes (mm³) from days 3−36 (**E**) and tumour growth rate analyses from days 15−36 (**F**) for xenograft models of VHH^HuR and T21RBCC-VHH^HuR (−/+ doxycycline). Blots shown in (**C**) are representative from n = 2 biologically independent samples per group. Data in (**E**) shows the mean and SD from n = 8 independent animals per group. Data in (**D**) show the individual data points and mean, or (**F**) mean and standard deviation from n = 2 and n = 8 biologically independent samples per group, respectively. Statistical significance was calculated using a two-way ANOVA and post-hoc test. Source data are provided as a Source Data file.

exemplifies the therapeutic opportunities afforded by targeted-protein degradation for benefit across a broad disease spectrum by targeting clinically relevant but hard-to-inhibit targets.

## Methods

### Ethical statement

These studies were conducted within the remit of a project licence approved by local Animal Welfare and Ethical Review Board (AWERB) committee and under a U.K. Home Office Project Licence in accordance with the U.K. Animals (Scientific Procedures) Act 1986 and EU Directive EU 2010/63/EU. Studies were performed according to the Home Office guidelines for the Care and Use of Laboratory Animals, and were also compliant with AstraZeneca policies on Bioethics and Good Statistical Practice in animal work.

### Expression and purification of recombinant HuR

pET24a vectors encoding HuR (full-length−AA 1–326; RNA-recognition motif (RRM) 1 and 2 (RRM1 + 2)−AA 11–186 and RRM1−AA 11–98) with

an Avi-tag and TEV-cleavable His-tag were used for protein expression in *Escherichia coli* BL21*λDE3 (New England Biolabs) following induction with a final concentration of 1 mM IPTG. Protein was purified by immobilised metal affinity chromatography (IMAC) and size-exclusion chromatography (SEC) with a HiLoad 16/600 Superdex 75 pg column (Cytiva), according to standard methods. Proteins were validated by SDS-PAGE and electrospray ionisation mass spectrometry (ESI-MS).

### Identification and confirmation of HuR-VHH-binding domains

HuR-specific-phage VHH were selected by performing three rounds of panning on biotinylated HuR protein (HuR-RNA-recognition motif 1 (RRM1) and RNA-recognition motif 2 (RRM2) (RRM1 + RRM2) or HuR RRM1 only) using a Llamda® phage display VHH library (composed of four sub-libraries)[75,76] built using Colibra® technology[75], and licenced from Isogenica − strategy outlined in Supplementary Data 8. Separately, a control Cas9 specific-phage-VHH was also isolated from the aforementioned Llamda® phage display VHH library by panning the full-length Cas9 (without bound RNA or DNA).

In brief, for each round of selection, bound phage-antigen complexes were captured on Dynabeads™ M-280 Streptavidin (Invitrogen), with specific-phage-VHH eluted in 10 μg/ml trypsin. *E. coli* TG1 were infected with eluted phage-VHH and plated on 2TY + ampicillin + glucose (2TYAG) bioassay plates. The following day, colonies were collected in 2TYA media and used to initiate the next round of panning by inoculating 2TYAG media. Cultures were infected with M13 helper phage prior to centrifugation and re-suspension of the pellet in 2TY + ampicillin + kanamycin (2TYAK) for overnight incubation. For the third round of selection, individual colonies were cultured in 2TYA and the following day deep-well blocks containing 2TYAG media were inoculated. IPTG was added at a final concentration of 1 mM for phage-VHH expression.

Cell pellets were collected via centrifugation and resuspended in BugBuster (Merck) prior to the transfer of soluble protein lysate supernatant (expressed phage VHH) to a fresh plate. MaxiSorp™ ELISA plates were prepared with 50 μl biotinylated HuR (RRM1 only or RRM1 + 2) (1–2 μg/ml) following standard protocol. 50 μl soluble protein lysate (expressed VHH) was added, followed by 50 μl rabbit anti-myc-HRP (Abcam) in 3% skimmed milk. Fifty microlitres 3,3′−5′5-tetramethylbenzidine (TMB) substrate (Merck) was then incubated for 2–5 min before the addition of 50 μl 0.5 M sulphuric acid. Optical density was measured at 450 nm using an EnVision plate reader (PerkinElmer). Positive HuR-VHH (VHH$^{HuR}$) binders were selected for sequencing analysis. Unique clones were prepared for larger scale expression and ELISA titrations against HuR RRM1.

## Plasmids for VHH characterisation and bioPROTAC evaluation

Twenty-four lead phage-VHH were cloned into a pFC14K FLAG-Halo-Tag® mammalian expression vector via PCR amplification and subsequent sub-cloning. VHH were located N-terminally to tags. For co-immunoprecipitations, the full-length HuR antigen was cloned upstream of emGFP in pTuner vector, as a regular mammalian expression vector by removing the cassette responsible for controlling expression. For use in NanoBRET, full-length HuR was cloned into the pFN31K Nluc CMV-neo Flexi® vector (Promega).

To generate the VHH-Fc fusion constructs, the VHH$^{GFP}$_4 sequence (Addgene plasmid #35579)[77] – as a non-HuR-targeting control−or VHH$^{HuR}$_17 were cloned alongside the hIgG1-Fc coding sequence from pFuse-hIgG1-Fc1 (InvivoGen) into the pGEM-HE vector giving rise to a pGEMHE-VHH-hIgG1-Fc1. A Fc mutant (H433A) that cannot bind TRIM21 was generated by site-directed mutagenesis (Agilent). For the bioPROTAC constructs, the sequence encoding the TRIM21 Ring-B-box-coiled-coil (T21RBCC) (AA 1-255) and VHH$^{HuR}$_17 or VHH$^{GFP}$[77] were integrated in a cassette with a T7 promoter containing AG initiator sequences, suitable UTR for use in in vitro mRNA transcription[78], Kozak consensus sequence and a C-terminal HA-tag. Constructs were generated in both an N- and C-orientation, or for the VHH$^{HuR}$ or VHH$^{GFP}$ domains alone. All cassettes were contained within a pcDNA 3.1(+) backbone (Invitrogen). For the T21RBCC$_{\Delta RING}$, sub-cloning was completed.

For stable inducible expression of bioPROTACs, the ObLiGaRe doxycycline-inducible (ODIn) system was utilised[48]. This system requires two vectors; a pZFN1-T2A-ZFN2-AAVS construct encoding two zinc finger nucleases (ZFN) targeting the AAVS locus and a pBSK construct housing the AAVS1 locus, Tet-On 3G inducible expression system, neomycin-resistance gene and cassette encoding the transgene (T21RBCC-VHH$^{HuR}$, T21RBCC-VHH$^{GFP}$ or VHH$^{HuR}$) followed by a T2A peptide sequence and mCherry reporter.

All constructs were confirmed via sequencing analysis.

## In vitro transcription

Prior to in vitro mRNA transcription, pcGEMHE or pcDNA 3.1 constructs were linearised via PCR 5′-capped modified RNA was synthesised according to the manufacturer's protocol using HiScribe™ T7 ARCA mRNA Kit (New England Biolabs) or HiScribe™ T7 High Yield RNA Synthesis Kit (New England Biolabs), CleanCap® (Trilink) and 5-Methoxyuridine-5′-Triphosphate (5-moUTP) (40% final concentration) (Trilink) respectively. mRNA was purified using the MEGAclear kit (ThermoFisher Scientific) prior to quality control on the 2100 Bioanalyzer (Agilent).

## Cell lines

The human HCT116 colorectal carcinoma cell line was obtained from the ECACC and the human A549 lung carcinoma, human U2OS osteosarcoma and human retinal pigment epithelial-1 (RPE-1) cell lines were obtained from the ATCC. All cell lines underwent short tandem repeat (STR) profiling prior to use and regular *Mycoplasma* screening. Cell lines were routinely passaged in DMEM medium (ThermoFIsher Scientific) supplemented with 10% foetal bovine serum (FBS), penicillin ($10^5$ U/L), and streptomycin (100 mg/L), and maintained at 37 °C and 5% $CO_2$ in a humidified environment.

## Transient transfection

For transient DNA expression, cells were transfected with FuGene HD transfection reagent (Promega) following the manufacturer's protocol.

For transient mRNA expression, cells underwent electroporation or chemical transfection. Electroporation was used delivery of mRNA encoding Fc-fusions and was performed using the Neon® Transfection System (ThermoFisher). Cells were washed with PBS and resuspended in Buffer R (ThermoFisher) at a concentration of $8 \times 10^7$ cells/ml. For each electroporation reaction $8 \times 10^5$ cells (10.5 μl) were mixed with 2 μl of antibody or mRNA or protein to be delivered (0.5 μM). This mixture was taken up into a 10 μl Neon® Pipette Tip (ThermoFisher) and electroporated using the following settings: 1400 V, 20 ms, 2 pulses. Electroporated cells were transferred to medium supplemented with 10% FCS without antibiotics. Transfection of T21RBCC constructs was undertaken via reverse transfection of mRNA (0.5 μg/ml) using Lipofectamine RNAiMAX transfection reagent (ThermoFisher Scientific) according to the manufacturer's protocol.

## Stable cell line generation

For cell line generation using the ODIn system[48], HCT116 cells were co-transfected with the ODIn vector (for either T21RBCC-VHH$^{HuR}$, T21RBCC-VHH$^{GFP}$ or VHH$^{HuR}$) and ZFN-AAVS vector and at a 2:1 ratio using FuGENE HD (Promega). For transgene selection integration, confluent cells were treated with G418 (500 μg/ml) (Sigma) for 10 days. For the induction of transgene expression, cells were treated with a final concentration of 100 ng/ml doxycycline for 24 h. Clonal selection was completed by fluorescence-activated cell sorting (FACS) of mCherry-expressing cells using the BD FACSAria™ II (BD Bioscience). Monoclonal populations were expanded and validated via immuno-blotting and immunofluorescence.

## MG132 treatment

Twenty-four hours post seeding, the HCT116 ODIn- T21RBCC-VHH$^{HuR}$ inducible cell line was co-treated with 5 μM MG132 and doxycycline (100 ng/ml) for 24 h.

## Co-immunoprecipitation

To confirm VHH-HuR interactions, the A549 cell line was seeded in T25 flasks and co-transfected with 150 ng VHH-FLAG-HaloTag® (lead VHH$^{HuR}$ clones or a VHH$^{Cas9}$ control) and emGFP-HuR. Forty-eight hours later, cells were lysed in NP40 lysis buffer, and supernatants were collected for subsequent quantification via a BCA assay (ThermoFisher Scientific). Two hundred and fifty micrograms cell lysate was prepared in 250 μl lysis buffer then incubated with 5 μl Protein G (for anti-FLAG IP) or Protein A (for anti-HuR IP) Dynabeads (ThermoFisher) for 1 h at room temperature. Lysates were collected and incubated with anti-

FLAG M2 Magnetic Beads (Sigma) or pre-prepared beads with anti-HuR antibody (Cell Signaling Technology) overnight at 4 °C. The following day, lysates were removed and beads washed with high salt wash buffer (1% NP40, 50 mM Tris HCl pH 7.5, 300 mM NaCl, 1 mM EGTA, 1 mM EDTA, 10 mM glycerophosphate, 50 mM sodium fluoride, 0.27 M sucrose, 5 mM sodium pyrophosphate, 1 mM sodium orthoVanadate). Bead-bound proteins were eluted in Laemmli buffer containing 10% β-mercaptoethanol. Lysate inputs and eluates were analysed via immunoblotting, as described below.

### NanoBRET

Further validation of the VHH$^{HuR\_8}$ and VHH$^{HuR\_17}$ interaction with HuR was completed using NanoBRET. HCT116 cells were seeded in 12-well plates and co-transfected with 0.5 ng HuR-NanoLuc (donor) and VHH-FLAG-HaloTag® (for VHH$^{HuR}$ clones or a VHH$^{Cas9}$ control) (acceptor), at a twofold serial dilution series, starting at 50 ng and a donor:acceptor ratio of 1:100. To maintain total DNA transfected, an empty pTuner vector was transfected for a total of 55 ng/condition. Twenty-four hours post transfection, a final concentration of 0.1 μM HaloTag® NanoBRET™ 618 Ligand (Promega) was added to cells. The following day, luciferase substrate was diluted in media (×166) and added for 10 min, then donor (450 mm) and acceptor (610 mm) emissions were measured using a luminometer.

### Determination of VHH$^{HuR}$ potency

Following induction with a final concentration of 1 mM IPTG, VHH$^{HuR\_8}$, VHH$^{HuR\_17}$ and VHH$^{Cas9}$ proteins were purified from *E. coli* TG1 by IMAC and SEC with a HiLoad 16/600 Superdex 75 pg column (Cytiva), according to standard methods. Proteins were validated by SDS-PAGE and peptide mass finger printing.

To determine the VHH$^{HuR}$ binding constants ($K_D$) to HuR, SPR was completed. A series S streptavidin Biacore chip (Cytiva) was docked into a T200 Biacore instrument (Cytiva), and priming was completed with running buffer (10 mM HEPES, 150 mM NaCl, 3 mM EDTA and 0.05% P-20, Cytiva). Biotinylated HuR RRM1, HuR RRM1 + 2 or full-length HuR protein (20 mg/ml) was injected for 420 s to give immobilisation signals of 1100, 1700 and 4800 respectively. Samples were prepared in assay buffer in a 384-well polypropylene microplate. Seven, threefold dilutions of VHH$^{HuR\_8/17/18}$, with a top concentration of 900 nM, were injected for 60 s with a dissociation time of 4000 s. All data were double referenced and globally fitted to a 1:1 binding model using the Biacore T200 Evaluation software.

To calculate VHH$^{HuR}$ inhibition constants ($K_i$), fluorescence polarisation assays were completed using a Musashi RNA-binding protein 1 (Msi1)-FITC probe (5′-rGrCrU rUrUrU rArArU rUrArU rUrUrU rG/3FluorT/− 3′). For confirmation of the probe $K_D$, full-length HuR protein (0–2000 nM) in the assay buffer (0.01 M HEPES pH 7.4, 0.15 M NaCl, 3 mM EDTA, 0.05% v/v Tween®20) and 10 nM Msi1-FITC probe were incubated in a 384-well plate and read immediately on a PHER-AStar plate reader. Probe $K_D$ was determined to be 0.0037 μM. To evaluate RNA competition, unlabelled RNA (5′-rGrCrU rUrUrU rArArU rUrArU rUrUrU rG-3′), VHH$^{HuR}$ (VHH$^{HuR\_8}$ or VHH$^{HuR\_17}$) or VHH$^{Cas9}$ (0–30 μM) were added to a plate followed by 8 nM Msi1-FITC then 8 nM HuR protein. Samples were incubated at room temperature for 5 min prior to reading on a PHERAStar plate reader. $K_i$ was calculated by forcing the fit to baseline and using a modified version of the Munson-Rodbard equation with probe $K_D$, as calculated above.

For confirmation of VHH$^{HuR\_17}$ inhibition of RNA binding, a series S streptavidin Biacore chip (Cytiva) was docked into an 8K Biacore instrument (Cytiva), and priming was completed as above. Biotinylated HuR RRM1 + 2 protein (20 mg/ml) was injected for 420 s. The A-B-A function of the 8K was used to measure RNA binding in the presence or absence of VHH$^{HuR\_17}$ by injecting 50 nM RNA (Musashi RNA-binding protein 1 (Msi1)-FITC probe (5′-rGrCrU rUrUrU rArArU rUrArU rUrUrU rG/3FluorT/-3′) (B) with 1 μM HuR$^{VHH\_17}$ as the flanking solution (A). 1 μM

of VHH$^{HuR\_17}$ was injected for 500 s prior to RNA to ensure equilibrium was obtained.

### Immunoblotting

Cells were seeded in 24-well plates then treated with mRNA or doxycycline (100 ng/ml). Following incubation, cells were washed in PBS and cell lysates were collected and prepared in 1x Laemmli sample buffer (BioRad) containing 10X Bolt™ Sample Reducing Agent containing 500 mM DTT (ThermoFisher Scientific). Samples were denatured at 95 °C prior to being resolved by SDS-PAGE and transferred onto PVDF membrane. Membranes were probed with antibodies outlined in the appendix, and proteins were detected by enhanced chemiluminescence (Amersham, GE Healthcare) and X-ray films or the GelDoc™ XR (BioRad), or visualised using an Odyssey DLx imaging system (LI-COR). Analyses were completed using ImageLab software (BioRad).

### Immunofluorescence

HCT116 cells were seeded in 384-well plates then transfected with mRNA encoding bioPROTACs. Following 18 h incubation, cells were washed in PBS, fixed for 15 min at room temperature in 4% methanol-free formaldehyde (ThermoFIsher Scientific) then blocked in 3% BSA and 0.1% Triton X-100 for 1 h. Cells were stained overnight with mouse anti-HuR (ThermoFisher Scientific) and rabbit anti-HA (Abcam) antibodies. Nuclei were stained with Hoechst (ThermoFisher Scientific) and the entire cell was stained with HCS CellMask™ Deep Red Stain (ThermoFisher Scientific). Cells were visualised using the CV7000 spinning disk confocal microscope (Yokogawa Inc.) using a 20× objective and 2 × 2 binning. Analyses were undertaken using Columbus software (PerkinElmer) to quantify HuR abundance. Due to some evidence of epitope competition between the VHH$^{HuR}$ and HuR antibody, immunofluorescence was only used as an orthogonal approach.

### Cell viability analysis

Parental and ODIn-HCT116 cell lines were prepared in phenol-red free DMEM medium (ThermoFisher Scientific) in a 384-well plate. After 24 h, cell lines were treated with a final concentration of 100 ng/ml doxycycline, then incubated for a further 72 h. Cell viability was assessed using the MTT (Sigma) cell assay according to the manufacturers protocol, and absorbance was measured at 570 nm using an EnVision plate reader (PerkinElmer).

### 3D colony formation assay

Parental and ODIn-HCT116 cell lines were prepared in 0.3% UltraPure low melting point agarose (ThermoFisher Scientific) diluted 1:1 in DMEM medium (ThermoFisher Scientific), and placed in 96-well plates pre-coated with 0.7% UltraPure low melting point agarose (ThermoFisher Scientific). Once set, DMEM supplemented with 500 μg/ml geneticin and 100 ng/ml doxycycline was added. Cells were grown for eight days, with regular media changes, then stained with SigmaFast BCIP/NBT solution (Sigma). Cells were visualised and colonies (size: 70–400 μm) were counted using the GelCount imager (Oxford Optronix).

### Proteomic sample preparation

For non-kinetic analyses, HCT116 cells were seeded in six-well plates and reverse transfected with mRNA encoding VHH$^{HuR}$, T21RBCC-VHH$^{HuR}$, VHH$^{HuR}$-T21RBCC, VHH$^{GFP}$, T21RBCC-VHH$^{GFP}$ and VHH$^{GFP}$-T21RBCC. For kinetic analyses, the HCT116 ODIn- T21RBCC-VHH$^{HuR}$ cell line was seeded in a 6-well plate 24 h prior to doxycycline induction at 24 h intervals over a 72-h period. At the endpoint, $1.5 \times 10^6$ cells/condition were collected following trypsinisation, and washing in cold PBS. Cell pellets were resuspended in S-Trap lysis buffer (5% SDS, 50 mM triethylammonium bicarbonate (TEAB) buffer, pH 7.55) and

solubilised using a Retsch mill (MM400) bead beater for 2 min at frequency 30 Hz. Protein concentration was measured using a BCA assay kit (Thermo Fisher). Fifty micrograms of protein lysates were digested using micro S-Trap method (Protifi.com) according to the manufacturer's protocol[79]. Proteins were reduced using 20 mM tris(2-carboxyethyl)phosphine for 15 min at 60 °C, alkylated using 80 mM iodoacetamide for 1 h at room temperature, and digested on a micro S-Trap cartridge using mass spectrometry grade trypsin/lys-C (Promega) for 2 h at 47 °C. Trypsin/Lys-C digested peptides were eluted with 50 mM TEAB buffer, followed by 0.2% formic acid (FA) in water, and 50/50 acetonitrile/water with 0.2% FA. Eluted peptides were dried then reconstituted in 0.15% FA in water.

## LC–MS/MS analysis

LC–MS/MS analysis was conducted on a timsTOF Pro mass spectrometer (Bruker) coupled with a nanoElute LC-system and nanoelectrospray ion source (CaptiveSpray Source, Bruker). Samples were loaded onto a 15 cm × 75 μm, 1.9 ReproSil, C18 column (PepSep.com) maintained at 50 °C. The peptides were separated using a gradient generated using solvent A (composed of 0.15% FA in water), and solvent B (composed of 0.15% FA in acetonitrile). Peptides were eluted at a flow rate of 500 nl/min over a 51 min gradient, from 4–24% solvent B (36 min), 24–36% solvent B (7 min), 36–64% solvent B (5 min), and 64–98% solvent B (3 min). Data-dependent acquisition (DDA) was performed in PASEF mode with six PASEF scans at a duty cycle close to 100%. MS acquisition recorded spectra from 100-1600 m/z and ion mobility was scanned from 0.85–1.30 Vs/cm$^2$ over a ramp time of 100 ms. The total duty cycle time was 1.15 s. The collision energy was linearly increased from 27 to 45 eV as a function of ion mobility. An active exclusion of 0.4 min was applied to precursors that reach a target intensity of 20,000 units. Data-independent acquisition (DIA)-PASEF mode was performed with a scheme that consists of two rows of 32 windows (eight PASEF scans per row and four steps per PASEF scan) with a 25 m/z isolation width[80]. The mass scan range was from 100 to 1700 m/z and ion mobility was scanned from 0.57–1.47 Vs/cm$^2$ over a ramp time of 100 ms. The collision energy was ramped linearly from 20 to 52 eV as a function of mobility.

## tims-TOF MS data analysis

To generate a comprehensive spectral library for the DIA analysis, we created a hybrid library that contained MS data of samples analysed in DDA mode and followed DIA analysis with technical replicates. The combined DDA and DIA acquisition raw files were analysed via Spectronaut (Biognosys AG) software with Pulsar search engine (SN14.10.201222) to build the library using UniProt human proteome database (UP000005640, 96,797 entries). The search parameters were set as default but included an additional deamidation (NQ) in variable modifications. DIA files (36 files for analysis of non-kinetic data and 20 files for analysis of kinetic data) were processed via Spectronaut using the default settings with precursor and protein FDR cut-off set to 0.01, quantification data filtering set to Q-value 0.5 percentile with global imputing, and cross run normalisation strategy set to local normalisation.

The mass spectrometry proteomics data have been deposited to the ProteomeXchange Consortium via the PRIDE[81] partner repository with the dataset identifiers PXD033221 (non-kinetic data) and PXD033222 (kinetic data).

## Proteomics data analyses

For the non-kinetic study, the lead bioPROTAC T21RBCC-VHH$^{HuR}$ and the controls VHH$^{HuR}$, VHH$^{GFP}$ and T21RBCC-VHH$^{GFP}$ underwent the outlined analyses. For the kinetic study all conditions were analysed. Peptide intensities were aggregated at the protein level. The resulting protein intensities were filtered and normalised to the total intensities of each sample. In-house scripts were developed to correct the batch effects using the plate information (https://github.com/AstraZeneca/trim21-bioprotac). To find differentially expressed proteins, a linear model was developed, and comparisons between constructs were defined using the Bioconductor limma package. False discovery rate (FDR)-adjusted p-values were calculated using the Benjamini Hochberg procedure and discoveries assigned based on a 5% FDR threshold and log$_2$FC < −1 (downregulation) or log$_2$FC > 1 (upregulation). Volcano plots and the Venn diagram were generated using the R packages ggplot2, ggrepel. Venn diagrams were generated using the R package VennDiagram. HuR protein-protein interaction information was retrieved from the BioGRID database (https://thebiogrid.org/). The web interface of UniprotKB (https://www.uniprot.org/) was used to extract the functional annotation (keywords and gene ontologies) of proteins present i.e. different subsets. Python scripts were used to process the output obtained from UniprotKB and barplots were generated using R (https://github.com/AstraZeneca/trim21-bioprotac).

## Xenograft model

Athymic nude female mice were obtained at 8 weeks of age from Envigo and housed in specific pathogen-free and standardised environmental conditions according to UK Home Office regulations. Mice received irradiated aspen chip bedding, nesting material, a cardboard tunnel, and wooden chew blocks. Mice were housed on a 12/12 light/dark cycle, with ad libitum UV-treated water and sterilised RM1 rodent diet. The maximum tumour burden was not allowed to exceed 10% of body weight, using the following formula based on calliper measurements of length(l) and width(w): volume = (pi/6)*l*w2. This maximum burden was not exceeded during this work. Sex was not considered in the study design, as we concluded for a human xenograft tumour model the sex of the recipient would not have a significant impact on the results seen.

For tumour engraftment, 8–12-week-old athymic nude mice were anaesthetised and injected subcutaneously in the flank with 5 × 10$^6$ HCT116-ODIn cell lines (20 mice for each of T21RBCC-VHH$^{HuR}$ or VHH$^{HuR}$ groups) in 100 μl sterile PBS. Russ Lenth's power tool was used to inform group sizes. Once tumours reached an average size of 150 mm$^3$, tumour-bearing mice were size-matched and randomly assigned into ten mice per experimental group prior to doxycycline treatment. For groups receiving doxycycline, mice were switched to a sterilised chow containing 625 ppm doxycycline hyclate (equivalent to 545 mg/kg doxycycline) (Ssniff). Throughout the duration of the study, tumour size was routinely measured with electronic calipers, enabling tumour volume to be calculated in mm$^3$ (length × width x width/2). Tumours were monitored until study endpoint or until an average tumour diameter of 15 mm or maximum volume of 1500 mm$^3$ was reached. At endpoint, mice were euthanised via cervical dislocation with secondary confirmation, and tumours were then resected for ex vivo analysis. The maximum tumour burden was not exceeded during this work (greater than 10% of body weight; determined using the following formula based on calliper measurements of length(l) and width(w): volume = (pi/6)*l*w2.

## In vivo imaging

Tumour-bearing mice were evaluated for mCherry expression at 4 days post doxycycline treatment by placing under recoverable isoflurane anaesthesia and imaging using an IVIS Spectrum (PerkinElmer) connected to XGI-8 Gas Anaesthesia System (Caliper Life Sciences). Once anaesthetised, mice were positioned on their sides on the IVIS stage enabling images to be captured with a field of view of 21.5 cm (FOV 'D'). Images were acquired by epiluminescence with excitation 587 nm and emission 610 nm for mCherry detection.

## Ex vivo analysis

Tumours were homogenised using zirconium oxide beads (1.4 mm and 2.8 mm) (Bertin Corp.) in PBS, containing protease/phosphatase

inhibitors (New England BioLabs) and benzonase nuclease (Merck), with the Precellys tissue homogeniser (6500 rpm, 3 × 30 s oscillations) (Bertin Technologies). Lysates were collected and prepared in 10× RIPA buffer (Merck), prior to protein quantification using the BCA protein assay kit (ThermoFisher Scientific). Tumour lysates were assessed by immunoblotting, as above.

## Statistical analyses

Tumour growth rate analysis was completed for each group from day 15 (first measurement post doxycycline) until day 36 (study endpoint), by fitting each animal's tumour volume data to an exponential model using equation '$log_{10}(tumour\ volume) = a + b * time + error$' where a and b are coefficients that correspond to the log initial volume and growth rate respectively, as previously described[82]. Growth rate summary metrics calculated for each animal were then used for statistical analysis to compare groups – treating each animal as the experimental unit.

Average (mean), standard deviation (s.d.) and statistical significance based on Student's t-test (two-tailed) or multiple comparisons using a one-way or two-way ANOVA with post-hoc test were calculated in GraphPad Prism.

## Reporting summary

Further information on research design is available in the Nature Portfolio Reporting Summary linked to this article.

# Data availability

Data supporting the findings of this study are presented within the article and supplemental information. Source data are provided within this manuscript, and the mass spectrometry proteomics data have been deposited to the ProteomeXchange Consortium via the PRIDE[81] partner repository with the dataset identifiers PXDO33221 (Degradation of HuR via TRIM21-based bioPROTAC – non-kinetic data) and PXDO33222 (Degradation of HuR via TRIM21-based bioPROTAC – kinetic data). Source data are provided with this paper.

# Material availability

Requests for resources and reagents should be directed to the lead contact James Hunt (james.hunt1@astrazeneca.com).

# Code availability

Code used for data and bioinformatics analyses is deposited on GitHub at https://github.com/AstraZeneca/trim21-bioprotac https://doi.org/10.5281/zenodo.8229693.

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

## Acknowledgements

This work was supported by the AstraZeneca R&D Postdoc programme and the MRC AstraZeneca Blue Sky programme. We also acknowledge contribution to this study from H. Olsson, Z. Rojnik for initial target discussions; L. Cazares, S. Hess, B. Zhang, for proteomic and mass spectrometry study support and experimental design; T. Tammsalu for creating in-house scripts for proteomics data analysis; S. Peel for assisting with imaging; A. Keen, R. McLeary for in vivo support; B. Phillips for advice on in vivo experimental design and data analysis; C. Bauer for feedback on the manuscript.

## Author contributions

A.F., D.C., L.J., J.H. conceived this project; A.F., D.C., E.d.V., T.M., F.M., J.T., E.W., G.D., P.R., E.G., J.D.T. completed in vitro experiments; S.M.C., R.C., J.W., A.C., S.H.S.W. completed and analysed proteomics experiments; T.M. performed xenograft experiments; all authors contributed to data analyses and interpretation; A.F. prepared the manuscript; all authors reviewed the manuscript; L.J. and J.H. supervised the project. All authors approved the final manuscript.

## Competing interests

A.F., S.M.C, E.dV., T.M., F.M., R.C., J.W., E.G., P.R., J.T., E.W., S.H.S.W., A.C., J.D.T., J.H. are present, or former employees and may be shareholders of AstraZeneca. The remaining authors declare no competing interests.
