## [Peer Review File · Nature Communications]

A TRIM21-based bioPROTAC highlights the therapeutic benefit of HuR degradationREVIEWER COMMENTS

Reviewer #1 (Remarks to the Author):

The manuscript by Fletcher et al. entitled "A novel TRIM21-based bioPROTAC highlights the therapeutic benefit of HuR degradation as an alternative for consideration for publication in Nature Communication.

The goal of this manuscript is twofold. The first goal is to report the development of the Trim-Away based bioPROTAC as a potential anti-cancer therapeutic agent with less off-target effects as previously developed ones. This sort of bioPROTAC has been previously reported by others (Chen et al., 2021 and Zeng et al., 2021) who tested it for its ability to degrade exogenous proteins (GFP and GFP tagged proteins). The authors of this current study show its ability to degrade endogenous protein for the first time. More importantly, the authors report an extensive array of assays, including proteomics data sets, that document the efficiency and specificity of this new tool. The LC- MS/MS data set is of particular interest to identify potential off-target effects of this new technique. The authors report that these off-target effects are minimal. While it may be so, they endeavor to publish the whole data set which will allow others with bioinformatic skills to mine for more information. The second goal was to investigate the ability of this new tool to target endogenous proteins such as HuR (aka ELAVL1) whose increased expression during certain cancers is suspected to play a role in its pathology. The authors hypothesized that the knock down of such a target might improve the outcome of cancer patients. Of course, this assumes the development of a cancer cell delivery system which is outside the scope of this study. However, the authors reports that the graft of one cancer cell line transduced with an inducible bioPROTAC into athymic mice. They show that the knockdown of HuR stalls the growth of tumor cells and therefore validates this new bioPROTAC technique.

Overall, it is a very exhaustive, worthy and, in my opinion, well-made study of the characterization and potential of this Trim-Way based bioPROTAC. While this study has potential for publication in Nature Communication, the HuR knockdown aspect of the story should be improved upon before publication.

Major points:

- It is not clear to me what the viability test shows in Figure 2E. The MTT detection kit is technically a marker of cellular metabolic activity, used as a surrogate of cytotoxicity, cell proliferation and/or cell death. However, I find this too vague, especially when we consider cancer cells which have unorthodox metabolic profiles. I do not believe these cells are undergoing cell death for a couple of reason (i.e. lack of tumor regression in the in vivo study and no decrease in expression of housekeeping genes in western blots (Figure 1K- vinculin, Fig 1I-GAPDH, Figure 2A-vinculin compare VHH HuR and T21RBCC-VHH Hu). So what is happening with these cells exactly? I think this study (and the future ones I trust) would benefit greatly if the authors could expand on what this viability means. Specifically, the authors should develop this further by investigating the effect of the HuR knock down on cell proliferation, arrest in specific cell phase markers of cell phases and cell death assay. Since the proteomics data show a decrease in IGFBP3 expression in HuR knockdown cells, the study would especially benefit from doing a proliferation assay.
- In my opinion, the in vivo results were underwhelming. The authors reported a stop of the tumor growth upon activation of their bioPROTAC construct, which was expected after all the data they had shown us in previous figures. However, the tumor did not shrink. Results are in line with lack of viability mentioned in figure 2E and their inability to form spheres in figure 2G, the mechanism of this staled tumor growth is still unexplained. This reiterates the need to further explore the mechanism of the cellular toxicity caused by a decrease in HuR expression. More importantly though, the authors did not report on the ability of these tumors to metastasize. They have chosen the colorectal cell line known for its high incidence of metastasis in the lung and the liver when injected in subQ in these mice. Have the authors looked at these tissues? Are Cherry positive cells present? This would give the manuscript more gravitas in terms of biological relevance. Provided the team can develop a targeted delivery of their bioPROTAC to tumors, this tool will appeal only to primary tumors that cannot be surgically removed. But if they show that their new tool also inhibits the

ability of the primary tumor to form metastasis, this now becomes a much more exciting venue.

Other points:

- Line 124: check out your sentence here. I believe you meant to put a period after "Properties".
- The authors use a lot of molecular based techniques that are not always well defined in the text, which makes the manuscript very hard to read for those not accustomed to these techniques. For example, in line 110, the authors mentioned the SPR (which I assume is the acronym for Surface Plasmon Resonance), but I could not find this acronym anywhere else, including in the material and method section "Determination of VHH HuR potency" which describe how these data were obtained. Same for "fluorescence polarization assay" on lane 114. I would also caution the authors to define better what NanoBRET is (e.g. a variation of FRET?).
- For figure 1: sometimes you spell out the cell line name on the panel and sometimes you don't. Please, homogenize your figure lay out.
- I did not find how the authors came about the phage display library they used to identify their HuR_8 and _17 . Please add the appropriate reference or create a supplemental material section to explain how this was made.
- Supplemental figure 1A. Please put a title on the figure and indicate which of the two partners was used at those various concentrations. It is not clear.
- In figure 1F: why are there 2 lanes for VHH HuR_17_Fc? If they are the same sample, one that has more signal for Fc than the other. Why? Please clarify.
- Figure 1I : why is there a size difference between T21RBCC-VHHHuR and VHHHuR-T21RBCC on the HA blot? Based on the author's explanation of these constructs, they are supposed to be the same molecule but with a different organization of its main components so I would expect them to be the same size.
- The authors mentioned in lane 211 that HuR is also defined as ELAVL1. This statement should be done in lane 205 when this topic was first broached.
- In lane 215-216, the author hypothesizes that there may be a nonspecific degradation of other ELAVL proteins (besides the intended target HuR) by their construct. This could be further substantiated by showing whether these ELAVL binds directly to Hub2 (CoIP or other technique). This experiment would greatly increase the impact of the paper as it gives an idea of the potential specificity of their degradation system.
- Lanes 225-230: the authors looked at the possible changes in proteomics profiles caused "specifically by the E3 ligase" part of their bioPROTAC and reported a decrease of YLPM1 and an increase of CCDC171 in their proteomics data. They do not comment about why YLPM1 decreases but they do hypothesize that the increase of CCDC171 could be linked to a putative multimer formation via the coiled-coil domain. This is dubious and does not make much sense unless the author thinks that this binding causes a stabilization of the protein as a result. I would either expand this section (and include what is known about YLPM1) or I would remove that sentence entirely.
- Simplify Figure 3: the authors chose to show their proteomic data using the aesthetically pleasing volcano plot and Ven diagram (which I generally do appreciate). However, the volcano plot will not be interactive in the manuscript and will be difficult to read and considering the format of the publication. The authors might be better off swapping some of those panels (specifically C through F) with a graph about data mentioned in the

manuscript (some of them currently in the supplemental data section). They can always put the volcano plot as supplemental material if wished.

· Supplemental figure 5I: please change the color or darken the shade of yellow used to create the lines on this graph, The light yellow is almost invisible. Also be aware that your y axis legend has been partially truncated during the figure formatting.

Reviewer #2 (Remarks to the Author):

Fletcher et al. report a TRIM21-based targeted protein degradation approach to mitigate the tumor-promoting effects of the RNA-binding protein Human antigen R (HuR). First, they use yeast display to identify a nanobody (VHH antibody) that inhibits RNA binding to HuR, and they then design a fusion of the best VHH-HuR candidate with TRIM21, with the N-terminal T21RBCC orientation providing optimal degradation results. They used a dox-induced system to study T21RBCC-VHH-HuR kinetics of bioPROTAC expression and HuR degradation, and they showed that degradation was dependent on the ubiquitin-proteasome system. T21RBCC-VHH-HuR treatment showed phenotypic changes in cell viability and colony formation, and the authors performed proteomic experiments to understand how T21RBCC-VHH-HuR might generate these effects. Finally, the authors show that T21RBCC-VHH-HuR-based depletion of HuR slows tumor growth in an in vivo xenograft HCT116 model. This is highly interesting work that has been conducted using well-planned experiments. I only have minor questions for the authors to address prior to publication.

1. Can the authors provide a bit more detail for their proteomics data analysis? For example, the authors state they used peptide intensities to filter for missing values and for normalization, which is not necessarily problematic, but these steps are often performed at the protein level. Did they indeed do their analyses at the peptide level before performing protein inference calculations to define protein groups? Can the authors also clarify what they mean by “duplicated instances” of peptides that were used for filtering, and what they mean by the median approach to intensity normalization. Finally, the in-house scripts (e.g., for mining Uniprot databases with python scripts) should be included as part of the supplemental information provided.

2. Did the authors consider a more statistically rigorous application of Uniprot keywords to understand functions of their differentially regulated proteins? Would something like a GO term analysis with background proteome correction provide any more insight into these hits, especially for the cases where >10 proteins are differentially regulated by T21RBCC-VHH-HuR treatment?

3. The human proteome database used for search database seems quite large (>96,000 entries). Can the authors comment on this, and can they explain if including various isoforms etc challenged data analysis when looking at protein groups with multiple members?

4. I commend the authors on cross-referencing hits from their proteomics data amongst three different controls, but would inclusion of any proteins that were differentially regulated in two of the three proteomics experiments (e.g., the 81 proteins in the Figure 3D Venn diagram between T21RBCC-VHH-HuR relative to VHH-HuR and VHH-GFP control groups) be warranted for understanding additional proteome alterations by T21RBCC-VHH-HuR? Focusing on the few hits they do make sense, but it seems there is more to understand by including other regulated proteins.

5. As the authors state, “the HCT116 cell line was selected owing to its demonstrable levels of HuR overexpression”, and all of the work shown in this manuscript is based on the

HCT116 model system. Can the authors show that an orthogonal cell line that also has shown HuR-dependence of phenotypic affects can be modulated by their approach?

Reviewer #3 (Remarks to the Author):

In the manuscript entitled "A novel TRIM21-based bioPROTAC highlights the therapeutic benefit of HuR degradation as an alternative to inhibition", Fletcher et al. engineered a TRIM21-based biological PROTAC that fused an identified VHH targeting HuR. This bioPROTAC induced degradation of endogenous HuR and displayed anti-tumorigenic effects. Identification novel antibodies against the clinically intractable pathogens or oncogenic proteins combined with ubiquitin-mediated target degradation provides a universal platform, which was demonstrated via TRIM21-based bioPROTAC to endogenous HuR in this manuscript. Here are the concerns to be addressed by authors,

1. In both in vitro and in vivo models, the bioPROTAC was electro-transfected into cell lines as mRNA. As the authors highlighted the therapeutic benefit of HuR degradation, please explain how to do in vivo delivery of bioPROTAC into cells in clinics.

2. P4. What does VHCas9 control refer to? The abbreviation that appears for the first time needs to be explained. And why was emGFP-HuR co-transfected alongside VHH-FLAG-HaloTag®?

3. Fig. 1F. Dose "control" mean no transfection or VHHHuR_17? There are two lines of VHHHuR_17-Fc, what's the difference? A more detailed description, like the dose of mRNA transfection, is needed here. Compared with control and VHHHuR_8-Fc, VHHHuR_17-Fc (H334A) also induced the degradation of HuR, which cannot draw this conclusion (P5 L133).

4. Fig. 2B. Why did ODIn system generation decrease the expression of HuR?

5. P6. In HCT116 cell line, targeted degradation of HuR decreases cell viability and colony formation. Authors should provide more evidence in other cancer and normal cells, which would solidify the versatility and safety of their approach.

6. As a therapeutic method, evaluating its safety in vivo and in vitro is necessary. P9. 'Importantly, no weight loss or adverse effects were observed as a result of doxycycline treatment or T21RBCC-VHHHuR expression.' This statement needs to be supported by experimental data. Moreover, the anti-tumorigenic effects were validated in mouse xenograft tumour models using the T21RBCC-VHHHuR ODIn cell lines. The targeting specificity of transfection via mRNA or DNA in vivo requires detection.

7. P13-14. 'For each electroporation reaction 8 x 10⁵ cells (10.5 µl) were mixed with 2 µl of antibody or mRNA or protein to be delivered.' The conc. in molarity should be expressed to understand what exactly is the active concentration. What are the dose and yield of mRNA transfections to degrade HuR in cells?

8. Fig. 1I. HuR degradation accompanied by the decrease of TRIM21 and antibodies, how to sustainably inhibit HuR in clinical application?

Minor points:

1. A schematic diagram of the mechanism is recommended to make it more intuitive and clearer for readers.

2. Some figures need to be uploaded in high resolution, such as Fig. 1I, 1K, 2A, 2C and 4C.

3. P25 L846. "VHHHuR_8/9/10/17 clones" should be "VHHHuR_8/9/18/17 clones".

4. Supplementary Figure 5M-Q are not mentioned in the manuscript.

Reviewer #4 (Remarks to the Author):

The authors provide a manuscript with novelty reporting the first case of TRIM21-based bioPROTAC targeting disease-relevant protein HuR, demonstrating that HuR would be a good target for future target-protein degradation campaign. A fusion attempt between the E3 TRIM21RBCC and the antibody VHHHuR demonstrates the feasibility of this novel bioPROTAC method. The authors did a good job on validating HuR degradation as a potential anticancer therapy. However, before considering for further publication, the following questions should be addressed.

Major:

Significance: Was there any attempt for small molecule PROTAC against HuR? If yes but failed, this could be the reason to turn to BioPROTAC. A.k.a, the advantage of choosing bioPROTAC instead of traditional small molecule PROTAC

About the HuB off-target identified in proteomics data, a binding data for HuB:VHHHuR could tell whether it is a HuR-HuB complex-mediated degradation or due to antibody-offtarget.

For the mouse experiment part, a HuR inhibition group should be set to evaluate the advantage of degradation over inhibition.

Minor:

A schematic illustration of the concept behind bioPROTAC will be helpful, as well as a ODIn-cassette design in the corresponding place.

Line 205: define the HuR synonym ELAVL1

Line 471: this should be "donor: acceptor ratio"

Line 265-267, FigS5A: Not only the T21RBCC-VHHGFP but also VHHHuR group shows TFAP4 decreasing. Does the TFAP4 change come from VHH side?

Fig1C: what is n.d.? I guess should be n.a. (not available) instead of n.d. (not detected) because clone 8 SPR competition was not performed.

FigS4A where does HA-tag come from?

Reviewer #1 (Remarks to the Author):

The manuscript by Fletcher et al. entitled “ A novel TRIM21-based bioPROTAC highlights the therapeutic benefit of HuR degradation as an alternative for consideration for publication in Nature Communication.

The goal of this manuscript is twofold. The first goal is to report the development of the Trim-Away based bioPROTAC as a potential anti-cancer therapeutic agent with less off-target effects as previously developed ones. This sort of bioPROTAC has been previously reported by others (Chen et al., 2021 and Zeng et al., 2021) who tested it for its ability to degrade exogenous proteins (GFP and GFP tagged proteins). The authors of this current study show its ability to degrade endogenous protein for the first time. More importantly, the authors report an extensive array of assays, including proteomics data sets, that document the efficiency and specificity of this new tool. The LC- MS/MS data set is of particular interest to identify potential off-target effects of this new technique. The authors report that these off-target effects are minimal. While it may be so, they endeavor to publish the whole data set which will allow others with bioinformatic skills to mine for more information. The second goal was to investigate the ability of this new tool to target endogenous proteins such as HuR (aka ELAVL1) whose increased expression during certain cancers is suspected to play a role in its pathology. The authors hypothesized that the knock down of such a target might improve the outcome of cancer patients. Of course, this assumes the development of a cancer cell delivery system which is outside the scope of this study. However, the authors reports that the graft of one cancer cell line transduced with an inducible bioPROTAC into athymic mice. They show that the knockdown of HuR stalls the growth of tumor cells and therefore validates this new bioPROTAC technique.

Overall, it is a very exhaustive, worthy and, in my opinion, well-made study of the characterization and potential of this Trim-Way based bioPROTAC. While this study has potential for publication in Nature Communication, the HuR knockdown aspect of the story should be improved upon before publication.

We thank the Reviewer for their thorough appraisal of our data and the constructive nature of their comments. We have now added further data and updated the manuscript in line with the Reviewer's comments and addressed their points below.

Major points:

1. It is not clear to me what the viability test shows in Figure 2E. The MTT detection kit is technically a marker of cellular metabolic activity, used as a surrogate of cytotoxicity, cell proliferation and/or cell death. However, I find this too vague, especially when we consider cancer cells which have unorthodox metabolic profiles. I do not believe these cells are undergoing cell death for a couple of reason (i.e. lack of tumor regression in the in vivo study and no decrease in expression of housekeeping genes in western blots (Figure 1K-vinculin, Fig 1I-GAPDH, Figure 2A-vinculin compare VHH HuR and T21RBCC-VHH Hu). So what is happening with these cells exactly? I think this study (and the future ones I trust) would benefit greatly if the authors could expand on what this viability means. Specifically, the authors should develop this further by investigating the effect of the HuR knock down on cell proliferation, arrest in specific cell phase markers of cell phases and cell death assay. Since the proteomics data show a decrease in IGFBP3 expression in HuR knockdown cells, the study would especially benefit from doing a proliferation assay.

In my opinion, the in vivo results were underwhelming. The authors reported a stop of the tumor growth upon activation of their bioPROTAC construct, which was expected after all the data they had shown us in previous figures. However, the tumor did not shrink. Results are in line with lack of viability mentioned in figure 2E and their inability to form spheres in figure 2G, the mechanism of this staled

tumor growth is still unexplained. This reiterates the need to further explore the mechanism of the cellular toxicity caused by a decrease in HuR expression.

We thank the Reviewer for their positive feedback on our TRIM21-based biological PROTAC data. Our data demonstrate the successful degradation of endogenous HuR and the subsequent biological effects this has owing to HuR overexpression being linked with high grade tumours and poor patient prognosis. As the Reviewer described, the MTT assay demonstrates measurable decrease in cellular metabolic activity which acts as a marker of cell viability, proliferation and cytotoxicity. Additionally, we have shown that bioPROTAC-mediated degradation of HuR caused a decrease in colony formation, which demonstrates an inability of these cells to proliferate (Figure 2F-G) in agreement with our observations made via the proteomics study (Franken N.A.P, et al. Clonogenic assay of cells in vitro. Molecular Cancer Therapeutics. 1, 2315-2319 (2006)). These in vitro observations resulted in an expected inhibition of tumour growth, which we believe to be of significant interest – particularly when looking at previous literature on the effects of bioPROTACs in xenograft tumour models (Ma, Y., et al. Targeted degradation of KRAS by an engineered ubiquitin ligase suppresses pancreatic cancer cell growth in vitro and in vivo. Molecular Cancer Therapeutics. 12, 286-294 (2013), Hatakeyama, S., et al. Targeted destruction of c-Myc by an engineered ubiquitin ligase suppresses cell transformation and tumor formation. Cancer Research. 65, 7874-7879 (2005)). However, we agree that further understanding of the mechanism underlying the effect on viability would be beneficial. Using samples retained from experiments described in figures 2A and 2B we have now added further Western blot data to assess cell cycle markers (See Supplementary Figure 4), these show a decrease in pHH3 and pCDK2, a result consistent with the observed reduction in proliferation being due to cell cycle disruption. Please also see text lines 200-205.

2. More importantly though, the authors did not report on the ability of these tumors to metastasize. They have chosen the colorectal cell line known for its high incidence of metastasis in the lung and the liver when injected in subQ in these mice. Have the authors looked at these tissues? Are Cherry positive cells present? This would give the manuscript more gravitas in terms of biological relevance. Provided the team can develop a targeted delivery of their bioPROTAC to tumors, this tool will appeal only to primary tumors that cannot be surgically removed. But if they show that their new tool also inhibits the ability of the primary tumor to form metastasis, this now becomes a much more exciting venue.

We thank the Reviewer for this interesting suggestion, as in addition to colorectal cancers having a high incidence of metastasis in the lung and liver upon subQ injection into mice, HuR has also been linked to metastatic disease (Dong, R., et al. An RNA-Binding Protein, Hu-antigen R, in Pancreatic Cancer Epithelial to Mesenchymal Transition, Metastasis, and Cancer Stem Cells. Molecular Cancer Therapeutics. 19, 2267-2277 (2020)). We agree with the Reviewer that looking at these tissues would be of interest, unfortunately these were not collected at the time of the experiment so retrospective analysis will not be possible. Regarding following mCherry positive cells, this is a challenge for in-life imaging, which could have been overcome by using a different reporter such as luciferase. We do not have such cell lines available to complete this work, which is also caveated by the ability to compare only those animals receiving chow containing doxycycline as the mCherry is only visible upon doxycycline induction. However, the Reviewer raises an interesting and important question and in life imaging using a different model could be investigated as part of future studies alongside tumour targeting approaches, for example those applied to AAV where HER2-AAV targets PD1 gene to Her2-RENCA tumour cells in BALB/c mice (Reul, J., et al. Tumor-Specific Delivery of Immune Checkpoint Inhibitors by Engineered AAV Vectors. Frontiers in Oncology. 9 (2019)). We have added text to the discussion to reflect this point (see line 347-351).

Other points:

3. Line 124: check out your sentence here. I believe you meant to put a period after “Properties”.

We have corrected this omission.

4. The authors use a lot of molecular based techniques that are not always well defined in the text, which makes the manuscript very hard to read for those not accustomed to these techniques. For example, in line 110, the authors mentioned the SPR (which I assume is the acronym for Surface Plasmon Resonance), but I could not find this acronym anywhere else, including in the material and method section “Determination of VHH HuR potency” which describe how these data were obtained. Same for “fluorescence polarization assay” on line 114. I would also caution the authors to define better what NanoBRET is (e.g. a variation of FRET?).

We thank the Reviewer for pointing these ambiguities out. We have now elaborated on these abbreviations and techniques.

5. For figure 1: sometimes you spell out the cell line name on the panel and sometimes you don't. Please, homogenize your figure lay out.

We apologise for these discrepancies and have now resolved this figure.

6. I did not find how the authors came about the phage display library they used to identify their HuR₈ and ₁₇. Please add the appropriate reference or create a supplemental material section to explain how this was made.

The phage library used was licensed from Isogenica and is described in the section “Identification and confirmation of HuR VHH-binding domains”. We have referenced all publicly available information, further details on the library would need to be obtained direct from Isogenica.

7. Supplemental figure 1A. Please put a title on the figure and indicate which of the two partners was used at those various concentrations. It is not clear.

We apologise to the Reviewer for this lack of clarity and have now addressed this to ensure the figure is clear.

8. In figure 1F: why are there 2 lanes for VHH HuR₁₇_Fc? If they are the same sample, one that has more signal for Fc than the other. Why? Please clarify.

We thank the Reviewer for pointing this ambiguity out. There are two lanes for the VHH HuR₁₇_Fc owing to the use of two different linkers in these constructs. At the time of this experiment, it was unclear if the linker would affect transient expression. We have adjusted the figure outlining the variation in these constructs, with additional information in the figure legend on these linkers.

9. Figure 1I : why is there a size difference between T21RBCC-VHHHuR and VHHHuR- T21RBCC on the HA blot? Based on the author's explanation of these constructs, they are supposed to be the same molecule but with a different organization of its main components so I would expect them to be the same size.

We thank the Reviewer for this comment. As the Reviewer describes, these constructs, and thus molecules, are the same but with the TRIM21 RBCC and VHH^{HuR} in different orientations. All constructs were confirmed via Sanger sequencing, we suspect that the subsequent proteins resolve at different speeds based on the protein charge and/or shape.

10. The authors mentioned in lane 211 that HuR is also defined as ELAVL1. This statement should be done in lane 205 when this topic was first broached.

We apologise for this oversight. The alternative name for HuR has now been reported when the subject was first mentioned.

11. In lane 215-216, the author hypothesizes that there may be a nonspecific degradation of other ELAVL proteins (besides the intended target HuR) by their construct. This could be further substantiated by showing whether these ELAVL binds directly to Hub2 (CoIP or other technique). This experiment would greatly increase the impact of the paper as it gives an idea of the potential specificity of their degradation system.

We thank the Reviewer for this insightful question. In the text, we have outlined how the literature describes an interaction between ELAVL2/HuB and ELAVL1/HuR (Hatanaka, T., et al., 2019, Uhlen, M., et al., 2017) and therefore we believe the co-degradation reflects the fact these two proteins form a native complex. Previously, co-degradation in Trim-away due to proteins forming complexes was described by Clift et al. (Clift, D. et al. A Method for the Acute and Rapid Degradation of Endogenous Proteins. Cell 171 (7), 1692-1706 (2017)). We have added text to reflect this point (see lines 227-228).

12. Lanes 225-230: the authors looked at the possible changes in proteomics profiles caused “specifically by the E3 ligase” part of their bioPROTAC and reported a decrease of YLPM1 and an increase of CCDC171 in their proteomics data. They do not comment about why YLPM1 decreases but they do hypothesize that the increase of CCDC171 could be linked to a putative multimer formation via the coiled-coil domain. This is dubious and does not make much sense unless the author thinks that this binding causes a stabilization of the protein as a result. I would either expand this section (and include what is known about YLPM1) or I would remove that sentence entirely.

We apologise to the Reviewer for this lack of clarity and have removed the aforementioned sentence.

13. Simplify Figure 3: the authors chose to show their proteomic data using the aesthetically pleasing volcano plot and Ven diagram (which I generally do appreciate). However, the volcano plot will not be interactive in the manuscript and will be difficult to read and considering the format of the publication. The authors might be better off swapping some of those panels (specifically C through F) with a graph about data mentioned in the manuscript (some of them currently in the supplemental data section). They can always put the volcano plot as supplemental material if wished.

We thank the Reviewer for the suggestion. We tried swapping the graphs as suggested but prefer to keep the current arrangement, if considered acceptable.

14. Supplemental figure 5I: please change the color or darken the shade of yellow used to create the lines on this graph, The light yellow is almost invisible. Also be aware that your y axis legend has been partially truncated during the figure formatting.

We apologise for this oversight and have modified the line colour on the graph and ensured the y-axis legend is fully legible.

Reviewer #2 (Remarks to the Author):

Fletcher et al. report a TRIM21-based targeted protein degradation approach to mitigate the tumor-promoting effects of the RNA-binding protein Human antigen R (HuR). First, they use yeast display to identify a nanobody (VHH antibody) that inhibits RNA binding to HuR, and they then design a fusion of the best VHH-HuR candidate with TRIM21, with the N-terminal T21RBCC orientation providing optimal degradation results. They used a dox-induced system to study T21RBCC-VHH-HuR kinetics of bioPROTAC expression and HuR degradation, and they showed that degradation was dependent on the ubiquitin-proteasome system. T21RBCC-VHH-HuR treatment showed phenotypic changes in cell viability and colony formation, and the authors performed proteomic experiments to understand how T21RBCC-VHH-HuR might generate these effects. Finally, the authors show that T21RBCC-VHH-HuR-based depletion of HuR slows tumor growth in an in vivo xenograft HCT116 model. This is highly interesting work that has been conducted using well-planned experiments. I only have minor questions for the authors to address prior to publication.

We thank the Reviewer for their thorough appraisal of our data and the constructive nature of their comments. We have now added further data and updated the manuscript in line with the Reviewer's comments and addressed their points below.

1. Can the authors provide a bit more detail for their proteomics data analysis? For example, the authors state they used peptide intensities to filter for missing values and for normalization, which is not necessarily problematic, but these steps are often performed at the protein level. Did they indeed do their analyses at the peptide level before performing protein inference calculations to define protein groups? Can the authors also clarify what they mean by “duplicated instances” of peptides that were used for filtering, and what they mean by the median approach to intensity normalization. Finally, the in-house scripts (e.g., for mining Uniprot databases with python scripts) should be included as part of the supplemental information provided.

We apologise to the Reviewer for omitting these details and provide clarity on these important questions below. Filtering and normalization were carried out at the protein level, not at the peptide level. Protein instances were duplicated in the protein output table generated from Spectronaut because we had chosen to include all peptides matched to each protein. Hence, duplicated protein instances needed to be filtered prior to subsequent analysis. Furthermore, we normalized protein intensities by total intensity, not median intensity. This means that we scaled protein intensities by dividing the intensity of each protein by the total intensity (i.e., the sum of all expression values) of the given sample. We have corrected the text and added more details about the proteomics data analysis in the Methods section. We have also updated our computer code and made all the code available on the GitHub page related to this manuscript (<https://github.com/AstraZeneca/trim21-bioprotac>), including the Python scripts used for mining the Uniprot databases.

2. Did the authors consider a more statistically rigorous application of Uniprot keywords to understand functions of their differentially regulated proteins? Would something like a GO term analysis with background proteome correction provide any more insight into these hits, especially for the cases where >10 proteins are differentially regulated by T21RBCC-VHH-HuR treatment?

Thank you to the reviewer for the suggested analysis. We have performed Gene Ontology (GO), KEGG pathway, Reactome, and WikiPathways enrichment analysis of the 71 up- and down-regulated proteins that were significantly altered at both 48 and 72 hours post induction using g:Profiler (<https://biit.cs.ut.ee/gprofiler>). Results show alteration of proteins involved in transmembrane and ion transport processes, which agrees with the Uniprot keyword analysis. We

have prepared a supplementary figure (S6.) that summarises the output and have referenced this in the text (see lines 265-267).

3. The human proteome database used for search database seems quite large (>96,000 entries). Can the authors comment on this, and can they explain if including various isoforms etc challenged data analysis when looking at protein groups with multiple members?

Apologies to the Reviewer for not providing this information. The database that we were using for this project included multiple protein isoforms and this is the reason we have >96,000 entries instead of ~20,000 entries. The Spectronaut software uses Parsimony principle and reports minimum possible number of protein identifications (protein groups) to assign all the peptides identified. If a peptide shared between isoforms A and B, that would lead to the grouping of A and B. If more peptides are identified for isoform A (other than the one peptide shared with B), Spectronaut only reports isoform A since probabilistically it is the one identified in your sample. The data we exported for the analysis was protein groups that already refined the isoform information.

4. I commend the authors on cross-referencing hits from their proteomics data amongst three different controls, but would inclusion of any proteins that were differentially regulated in two of the three proteomics experiments (e.g., the 81 proteins in the Figure 3D Venn diagram between T21RBC-VHH-HuR relative to VHH-HuR and VHH-GFP control groups) be warranted for understanding additional proteome alterations by T21RBC-VHH-HuR? Focusing on the few hits they do make sense, but it seems there is more to understand by including other regulated proteins.

Thank you to the Reviewer for the suggested analysis. To obtain an unbiased understanding of the additional proteome alterations caused by T21RBC-VHH-HuR, we have performed a Gene Ontology (GO), KEGG pathway, Reactome and WikiPathways of the 7, 81, 88 and 90 proteins highlighted in the Venn diagram in Fig. 3D using g:Profiler (<https://biit.cs.ut.ee/qprofiler/gost>).

The central subset of 7 proteins shows enrichment of mRNA 3'-UTR binding activity as 3 out of the 7 proteins (ELAVL1, ELAVL2 and IGF2BP3) are annotated with this molecular function term.

ID	Source	Term ID	Term Name	P _{adj} (query_1)
1	GO:MF	GO:0003730	mRNA 3'-UTR binding	6.596×10 ⁻⁴
2	GO:CC	GO:0010494	cytoplasmic stress granule	4.777×10 ⁻²

GO:MF		stats				ELAVL1	ELAVL2	IGF2BP3	TFAP4	SDCA	RNF115	B4GALT5
Term name	Term ID	P _{adj}	$-\log_{10}(P_{adj})$	0	≤16							
mRNA 3'-UTR binding	GO:0003730	6.596×10 ⁻⁴										
mRNA binding	GO:0003729	2.868×10 ⁻²										

1 to 2 of << Page 1 of 1 >>

GO:CC		stats				ELAVL1	ELAVL2	IGF2BP3	TFAP4	SDCA	RNF115	B4GALT5
Term name	Term ID	P _{adj}	$-\log_{10}(P_{adj})$	0	≤16							
cytoplasmic stress granule	GO:0010494	4.777×10 ⁻²										

Looking now at downregulated proteins under at least two out of the three conditions, the subset of 81 proteins in Fig. 3D shows enrichment of the biological processes involving mRNA and nucleocytoplasmic transport. Also, more than half of the proteins degraded are annotated to be co-localised in the nucleoplasm and the nuclear pore.

ID	Source	Term ID	Term Name	P _{adj} (query_1)
1	GO:MF	GO:0005515	protein binding	1.518×10 ⁻³
2	GO:MF	GO:0017056	structural constituent of nuclear pore	1.725×10 ⁻³
3	GO:MF	GO:0003677	DNA binding	4.557×10 ⁻²
4	GO:BP	GO:0051028	mRNA transport	5.981×10 ⁻⁵
5	GO:BP	GO:0006913	nucleocytoplasmic transport	8.021×10 ⁻⁵
6	GO:BP	GO:0008104	protein localization	4.536×10 ⁻⁴
7	GO:BP	GO:0006996	organelle organization	1.113×10 ⁻³
8	GO:BP	GO:0080090	regulation of primary metabolic process	1.144×10 ⁻³
9	GO:BP	GO:0051292	nuclear pore complex assembly	8.317×10 ⁻³
10	GO:BP	GO:0007049	cell cycle	1.668×10 ⁻²
11	GO:CC	GO:0005654	nucleoplasm	5.822×10 ⁻¹⁹
12	GO:CC	GO:0005643	nuclear pore	7.011×10 ⁻⁷
13	GO:CC	GO:0015630	microtubule cytoskeleton	6.491×10 ⁻³

The functional enrichments observed in the aggregated subsets of 88 (7+81) and 90 (7+81+1+1) proteins are similar to those presented in the subset of 81 proteins alone.

Subset of 88:

ID	Source	Term ID	Term Name	P _{adj} (query_1)
1	GO:MF	GO:0005515	protein binding	1.227×10 ⁻³
2	GO:MF	GO:0017056	structural constituent of nuclear pore	2.200×10 ⁻³
3	GO:BP	GO:0051028	mRNA transport	6.406×10 ⁻⁶
4	GO:BP	GO:0006913	nucleocytoplasmic transport	2.041×10 ⁻⁵
5	GO:BP	GO:0080090	regulation of primary metabolic process	9.774×10 ⁻⁵
6	GO:BP	GO:0008104	protein localization	1.019×10 ⁻³
7	GO:BP	GO:0006996	organelle organization	3.473×10 ⁻³
8	GO:BP	GO:0051292	nuclear pore complex assembly	1.168×10 ⁻²
9	GO:BP	GO:0007049	cell cycle	2.001×10 ⁻²
10	GO:CC	GO:0005654	nucleoplasm	2.537×10 ⁻¹⁰
11	GO:CC	GO:0005643	nuclear pore	1.519×10 ⁻⁶
12	GO:CC	GO:0015630	microtubule cytoskeleton	2.106×10 ⁻²

Subset of 90:

ID	Source	Term ID	Term Name	P _{adj} (query_1)
1	GO:MF	GO:0005515	protein binding	8.071×10 ⁻⁴
2	GO:MF	GO:0017056	structural constituent of nuclear pore	2.587×10 ⁻³
3	GO:MF	GO:0003677	DNA binding	3.412×10 ⁻²
4	GO:BP	GO:0051028	mRNA transport	8.627×10 ⁻⁶
5	GO:BP	GO:0006913	nucleocytoplasmic transport	2.920×10 ⁻⁵
6	GO:BP	GO:0080090	regulation of primary metabolic process	8.616×10 ⁻⁵
7	GO:BP	GO:0008104	protein localization	5.424×10 ⁻⁴
8	GO:BP	GO:0006996	organelle organization	2.208×10 ⁻³
9	GO:BP	GO:0007049	cell cycle	8.933×10 ⁻³
10	GO:BP	GO:0051292	nuclear pore complex assembly	1.371×10 ⁻²
11	GO:CC	GO:0005654	nucleoplasm	3.127×10 ⁻¹¹
12	GO:CC	GO:0005643	nuclear pore	1.881×10 ⁻⁶
13	GO:CC	GO:0015630	microtubule cytoskeleton	2.888×10 ⁻²

Overall, the core set of 7 proteins downregulated in all the conditions is enriched for mRNA 3'-UTR binding activity however when extending the set to all proteins downregulated in at least two of the conditions additional functions emerge, namely proteins involved in mRNA and nucleocytoplasmic transport, as well enrichment in the nucleoplasm and nuclear pore.

5. As the authors state, “the HCT116 cell line was selected owing to its demonstrable levels of HuR overexpression”, and all of the work shown in this manuscript is based on the HCT116 model system. Can the authors show that an orthogonal cell line that also has shown HuR-dependence of phenotypic affects can be modulated by their approach?

We thank the Reviewer for raising this important query. As the Reviewer highlights, HuR degradation decreases both cell viability and proliferation in the HCT116 cell line both in vitro and

in vivo (Figure 2 and Figure 4) however, despite showing the ability of our TRIM21 bioPROTAC to degrade HuR in both the A549 and U2OS cell lines (Supplementary figure 2) there is no biological effect data in these cell lines within this investigation. It has previously been described that nanoparticles containing HuR-targeting siRNA were able to significantly decrease HuR expression, inhibit invasion and migration and induce apoptosis in A549 cells (Muralidharan, R. et al. Tumor-targeted nanoparticle delivery of HuR siRNA inhibits lung tumor growth in vitro and in vivo by disrupting the oncogenic activity of the RNA-binding protein HuR. *Molecular Cancer Therapeutics* 16, 1470-1486 (2017)). With respect to the U2OS osteosarcoma cell line, HuR is known to drive disease progression (Li, Z., et al. LncRNA B4GALT1- AS1 recruits HuR to promote osteosarcoma cells stemness and migration via enhancing YAP transcriptional activity. *Cell Proliferation*, 1-11 (2018)). To evaluate the phenotypic effect of HuR degradation we had to engineer doxycycline-inducible bioPROTAC cell lines, which means that we cannot replicate the phenotype without generating another cell line. This is a lengthy and labour-intensive process. We hope the Reviewer accepts that we feel this is beyond the scope of the current study, but that given previous observations regarding HuR dependence we would expect the phenotype observed to be replicated across other cancer lines that have previously been shown to have an HuR dependency.

Reviewer #3 (Remarks to the Author):

In the manuscript entitled “A novel TRIM21-based bioPROTAC highlights the therapeutic benefit of HuR degradation as an alternative to inhibition”, Fletcher et al. engineered a TRIM21-based biological PROTAC that fused an identified VHH targeting HuR. This bioPROTAC induced degradation of endogenous HuR and displayed anti-tumorigenic effects. Identification novel antibodies against the clinically intractable pathogens or oncogenic proteins combined with ubiquitin-mediated target degradation provides a universal platform, which was demonstrated via TRIM21-based bioPROTAC to endogenous HuR in this manuscript. Here are the concerns to be addressed by authors,

We thank the Reviewer for their thorough appraisal of our data and the constructive nature of their comments. We have now added further data and updated the manuscript in line with the Reviewer's comments and addressed their points below.

1. In both in vitro and in vivo models, the bioPROTAC was electro-transfected into cell lines as mRNA. As the authors highlighted the therapeutic benefit of HuR degradation, please explain how to do in vivo delivery of bioPROTAC into cells in clinics.

We thank the reviewer for raising this important point. In a previous study we explored the delivery of intracellular biologics using both mRNA and AAV (De Genst, E., et al. Blocking phospholamban with VHH intrabodies enhances contractility and relaxation in heart failure. Nature Communications 13 (1), 1-13 (2022)). Specifically in the case of cancer, AAV is now being used to target tumours in pre-clinical models (we have added text to reflect this, see lines 347-351), providing a possible route forward to clinical application, albeit at an early stage of development.

2. P4. What does VHHCas9 control refer to? The abbreviation that appears for the first time needs to be explained. And why was emGFP-HuR co-transfected alongside VHH-FLAG-HaloTag®?

We thank the Reviewer for pointing this ambiguity out. The VHH^{Cas9} control is a VHH control, which binds to the Cas9 protein. As the cells used in this investigation do not express Cas9, this is a negative control VHH. The text has been updated to address this omission. With respect to the co-immunoprecipitation - to enhance the success of this assay, the exogenous emGFP-HuR was transfected into cells to ensure that identification of the VHH-HuR interaction was possible. Figure 1A highlights the interaction between VHH^{HuR-8} and VHH^{HuR-17} with both the endogenous HuR as well as emGFP-HuR however, in the case of the VHH^{HuR-8} interaction with HuR this is more difficult to observe via the Western blot, but can be confirmed via the overexpressed emGFP-HuR.

3. Fig. 1F. Dose “control” mean no transfection or VHHHuR_17? There are two lines of VHHHuR_17-Fc, what's the difference? A more detailed description, like the dose of mRNA transfection, is needed here. Compared with control and VHHHuR_8-Fc, VHHHuR_17-Fc (H334A) also induced the degradation of HuR, which cannot draw this conclusion (P5 L133).

We thank the Reviewer for pointing these ambiguities out. All mRNA were resuspended in water, with the control being delivery of water alone. There are two lanes for the VHH HuR_17_Fc owing to the use of two different linkers in these constructs. At the time of this experiment, it was unclear if the linker would affect transient expression. We have adjusted the figure outlining the variation in these constructs, with additional information in the figure legend on these linkers. mRNAs were delivered at an electroporation tip concentration of 80nM. We have adjusted the text on lines 135-137 to more accurately reflect the observations in Figure 1F. TRIM21 recruitment to the same construct without the H433A mutation causes a significant increase in degradation; however, we agree with the Reviewer that the H334A mutation, although effective, does not appear to

completely return the levels of HuR to those of the control shown in Figure 1F. The mechanism for this is unclear, the Fc-VHH fusion is a dimer in the Fc format which could lead to an avidity effect with respect to HuR binding and therefore have a minor effect on HuR stability, but extensive investigation of manifold constructs transfected into HCT116 cells showed no significant effect on HuR levels (see below). Nonetheless as we subsequently used, and extensively characterised, the RBCC-VHH format we did not investigate potential TRIM21-independent functions of the VHH_Fc format further.

Figure: Western blot analysis in HCT116 cells transfected with mRNA encoding bivalent or trivalent HuR VHH (n=3)

4. Fig. 2B. Why did ODI system generation decrease the expression of HuR?

We thank the Reviewer for highlighting this. ODI cell lines were generated via transfection with an ODI and ZFN-AAVS vectors prior to selection with G418. Clonal selection was then completed for monoclonal populations for each construct. It is possible that this process or clonal selection has resulted in slight variations in terms of HuR expression compared to the parental cell lines. Critically, all inducible cell lines show comparable expression levels of HuR.

5. P6. In HCT116 cell line, targeted degradation of HuR decreases cell viability and colony formation. Authors should provide more evidence in other cancer and normal cells, which would solidify the versatility and safety of their approach.

We thank the Reviewer for raising this important query. Our data clearly demonstrates that HuR degradation decreases both cell viability and proliferation in the HCT116 cell line both in vitro and in vivo (Figure 2 and figure 4) however, despite showing the ability of our TRIM21 bioPROTAC to degrade HuR in both the A549 and U2OS cell lines (Supplementary figure 2) there is no biological effect data in these cell lines. It has previously been described that nanoparticles containing HuR-targeting siRNA were able to significantly decrease HuR expression, inhibit invasion and migration and induce apoptosis in A549 cells (Muralidharan, R. et al. Tumor-targeted nanoparticle delivery of HuR siRNA inhibits lung tumor growth in vitro and in vivo by disrupting the oncogenic activity of the RNA-binding protein HuR. *Molecular Cancer Therapeutics* 16, 1470-1486 (2017)). With respect to the U2OS osteosarcoma cell line, HuR is known to drive disease progression (Li, Z., et al. LncRNA B4GALT1- AS1 recruits HuR to promote osteosarcoma cells stemness and migration via enhancing YAP transcriptional activity. *Cell Proliferation*, 1-11 (2018)). To evaluate the phenotypic effect of HuR degradation we had to engineer doxycycline-inducible bioPROTAC cell lines, which means that we cannot replicate the phenotype without generating another cell line. We hope the Reviewer accepts

that we feel this is beyond the scope of the current study, but that given previous observations regarding HuR dependence we would expect the phenotype observed to be replicated across other cancer lines that have previously been shown to have an HuR dependency.

6. As a therapeutic method, evaluating its safety in vivo and in vitro is necessary. P9. 'Importantly, no weight loss or adverse effects were observed as a result of doxycycline treatment or T21RBCC-VHHHuR expression.' This statement needs to be supported by experimental data. Moreover, the anti-tumorigenic effects were validated in mouse xenograft tumour models using the T21RBCC-VHHHuR ODIn cell lines. The targeting specificity of transfection via mRNA or DNA in vivo requires detection.

Thank you to the Reviewer for highlighting this omission. We have now included these data in the supplementary material (Supplementary figure 8I). These data demonstrate there was no weight loss because of doxycycline treatment or T21RBCC-VHH^{HuR-17} expression. However, some weight loss was observed in these groups at either a lower or similar extent to the non-treated groups and is likely due to the tumour burden. With respect to targeting specificity of our bioPROTAC, in this xenograft model using the ODIn cell lines, mCherry was used as a reporter for detection of the T21RBCC-VHH^{HuR} (Supplementary figure 8A-B, D). Having successfully observed an anti-tumour effect in these xenograft studies, we now feel that future studies can focus on the targeted delivery of a TRIM21 bioPROTAC as mRNA/DNA.

7. P13-14. 'For each electroporation reaction 8 x 10⁵ cells (10.5 µl) were mixed with 2 µl of antibody or mRNA or protein to be delivered.' The conc. in molarity should be expressed to understand what exactly is the active concentration. What are the dose and yield of mRNA transfections to degrade HuR in cells?

We thank the Reviewer for highlighting these omissions – the concentration of mRNA has been updated in the methods section. 10.5ul cells were electroporated with 2ul of mRNA at a concentration of 0.5uM, resulting in a final electroporation tip concentration of 80nM.

8. Fig. 1I. HuR degradation accompanied by the decrease of TRIM21 and antibodies, how to sustainably inhibit HuR in clinical application?

We thank the Reviewer for raising this important point - please see the reply to this incorporated at question 1.

Minor points:

9. A schematic diagram of the mechanism is recommended to make it more intuitive and clearer for readers.

Thanks to the Reviewer for this suggestion, we have now included a schematic for the reader.

10. Some figures need to be uploaded in high resolution, such as Fig. 1I, 1K, 2A, 2C and 4C.

Thank you to the Reviewer for this feedback. These figures are in the highest resolution possible because of data being obtained at this resolution on the ChemiDoc. All raw data can also be found in the source data file.

11. P25 L846. "VHHHuR 8/9/10/17 clones" should be "VHHHuR 8/9/18/17 clones".

Thank you to the Reviewer for noting this typographical error, we have corrected this.

12. Supplementary Figure 5M-Q are not mentioned in the manuscript.

We apologise for this oversight and have now referenced these figures.

Reviewer #4 (Remarks to the Author):

The authors provide a manuscript with novelty reporting the first case of TRIM21-based bioPROTAC targeting disease-relevant protein HuR, demonstrating that HuR would be a good target for future target-protein degradation campaign. A fusion attempt between the E3 TRIM21RBCC and the antibody VHHHuR demonstrates the feasibility of this novel bioPROTAC method. The authors did a good job on validating HuR degradation as a potential anticancer therapy. However, before considering for further publication, the following questions should be addressed.

We thank the Reviewer for their thorough appraisal of our data and the constructive nature of their comments. We have now added further data and updated the manuscript in line with the Reviewer's comments and addressed their points below.

Major:

1. Significance: Was there any attempt for small molecule PROTAC against HuR? If yes but failed, this could be the reason to turn to BioPROTAC. A.k.a, the advantage of choosing bioPROTAC instead of traditional small molecule PROTAC.

Thank you to the reviewer for raising this point. In assessing HuR as a target for validation in the context of targeted protein degradation there was no literature small molecule with an appropriate profile to generate a tool low molecular weight PROTAC. Producing such a molecule would require significant efforts in screening and medicinal chemistry. We therefore looked towards the opportunity to rapidly generate an intracellular biologic that could then be turned into a bioPROTAC, for the purpose of comparing inhibition to degradation as a route to modulating HuR in a cancer setting. We hope this study provides encouragement for further effort to identify small molecule PROTACs for the treatment of cancer, or perhaps further exploration of the opportunity afforded by bioPROTACs.

2. About the HuB off-target identified in proteomics data, a binding data for HuB:VHHHuR could tell whether it is a HuR-HuB complex-mediated degradation or due to antibody-off-target.

We thank the Reviewer for this insightful question. In the text, we have outlined how the literature describes an interaction between ELAVL2/HuB and ELAVL1/HuR (Hatanaka, T., et al., 2019, Uhlen, M., et al., 2017) and therefore we believe the co-degradation reflects the fact these two proteins form a native complex. Previously, co-degradation in Trim-away due to proteins forming complexes was described by Clift et al. (Clift, D. et al. A Method for the Acute and Rapid Degradation of Endogenous Proteins. Cell 171 (7), 1692-1706 (2017)). We have added text to reflect this point (see lines 227-228). We believe that these literature observations, and our global proteomic dataset, provide strong evidence for the specificity of our approach.

3. For the mouse experiment part, a HuR inhibition group should be set to evaluate the advantage of degradation over inhibition.

We apologise for any confusion in our results, the VHH^{HuR} is a HuR inhibitor, which in this investigation was demonstrated to bind HuR with low nanomolar affinity and inhibit HuR binding to RNA. Therefore, the VHH^{HuR} was included in our study to enable the evaluation of degradation over inhibition. Figure 2 and Figure S8 demonstrate that this VHH^{HuR} inhibitor does not have a biological effect either in vitro or in vivo, in contrast to the T21RBCC-VHH^{HuR} bioPROTAC which negatively impacts cell viability and proliferation as well as tumour growth in vivo. Therefore, we conclude that there is a clear advantage of HuR degradation over inhibition.

Minor:

4. A schematic illustration of the concept behind bioPROTAC will be helpful, as well as a ODIn-cassette design in the corresponding place.

Thanks to the Reviewer for this suggestion, we have now included a graphical abstract of the bioPROTACs concept for the reader. As outlined in the text, ODIn cassettes were designed for dual expression of the bioPROTAC (Figure 1H) alongside an mCherry reporter, following the cassette design elegantly reported in Lundin, A., et al., 2020. We hope the reference provided is sufficient for the reader to obtain a detailed understanding of the nature of the constructs used.

5. Line 205: define the HuR synonym ELAVL1

ELAVL1 has been defined upon first mention within the introduction.

6. Line 471: this should be “donor: acceptor ratio”

Thanks to the Reviewer for highlighting this error. We have now corrected this.

7. Line 265-267, FigS5A: Not only the T21RBCC-VHHGFP but also VHHHuR group shows TFAP4 decreasing. Does the TFAP4 change come from VHH side?

Thanks to the Reviewer for highlighting this interesting point. Due to the ability of VHH^{HuR} to inhibit HuR, it is possible that this effect on TFAP4 could be observed via both HuR inhibition and degradation. However, in agreement with the other observations made it is evident that HuR degradation results in a more apparent effect on TFAP4. Further, any effect of the VHH^{HuR} on TFAP4 does not translate to a biological effect. Future studies investigating these proteins identified via proteomics may prove to be insightful in answering such questions.

8. Fig1C: what is n.d.? I guess should be n.a. (not available) instead of n.d. (not detected) because clone 8 SPR competition was not performed.

We apologise for this oversight and have now corrected this.

9. FigS4A where does HA-tag come from?

Many thanks to the Reviewer for this question. The bioPROTACs transfected into the A549 and U2OS cell lines shown in Supplementary Figure 4 contain a HA-tag similarly to the bioPROTACs transfected into the HCT116 cell line (Figure 1I). The HA tag on these bioPROTACs is outlined in paragraph 3 of section ‘A TRIM21-based bioPROTAC degrades HuR’.

REVIEWERS' COMMENTS

Reviewer #1 (Remarks to the Author):

I have carefully reviewed the authors responses to my review and the subsequent changes they did to their manuscripts.

The majority of my comments were adequately addressed.

Some of my comments could not be addressed at this point for reasons that were very compelling (e.g. point 2- study of pulmonary metastasis in vivo) and I accept their arguments.

Other comments were partially answered to my satisfaction. For point 1, for example, I requested the authors went deeper into the apparent cellular toxicity caused by HuR degradation. I specifically suggested they probe cell cycle markers and did show that M phase and S and G2 markers were indeed decreased in HuR trimmed cells, substantiating their claim that HuR knock down cells have impaired cell proliferation and therefore can be used as a tool to stall tumor growth in vivo. They did not perform the cell death assay I requested, however. While the apparently unchanged levels of housekeeping proteins such as GAPDH and Vinculin argued against it, it was by no mean a definitive proof. It is disappointing that the authors chose not to push the cellular toxicity a bit further but, as is, it should be sufficient for this publication. However, I advise the authors to look into that more substantially, especially if they push this system for clinical trials. The inadvertent delivery of HuR-BioPROTAC into healthy cells could cause serious issues (especially if Apoptosis is triggered) either as a results of HuR degradation directly or the degradation of some of its putative partners (ELAVL 2, IGF-BP3, TF-AP4) or putative HuR-bioPROTAC cross-reactants (ELAVL 2). Anticipating their root cause of these issues could prove valuable.

In one instance (item 14), the authors claimed to have introduced the requested change but did not. Supplemental figure 7 I (former supplemental figure 5I) was made using shades of yellow too pale to see. They are still the same color. I will leave it to the editor to make that decision.

Overall, the authors have delivered an impressive amount of data on the development of a new tool to combat cancer. They made a fair attempt at answering my questions and concerns and the resulting manuscript is, in my opinion, fit enough for publication.

Reviewer #2 (Remarks to the Author):

The authors have done well to address my comments. That mass spectrometry data analysis section is now clearer, and the additional information about various downregulated proteins in two out of three conditions is interesting and shows trends congruent with their original data. I can appreciate the challenges of generating a new doxycycline-inducible bioPROTAC cell line, and agree that applicability to other systems can be better addressed in follow up studies. With that, I see this manuscript as suitable for publication.

Reviewer #3 (Remarks to the Author):

The authors have addressed my concerns.

Reviewer #4 (Remarks to the Author):

The authors have addressed all my previous concerns.

Reviewer #1 (Remarks to the Author):

I have carefully reviewed the authors responses to my review and the subsequent changes they did to their manuscripts.

The majority of my comments were adequately addressed.

Some of my comments could not be addressed at this point for reasons that were very compelling (e.g. point 2- study of pulmonary metastasis in vivo) and I accept their arguments.

Other comments were partially answered to my satisfaction. For point 1, for example, I requested the authors went deeper into the apparent cellular toxicity caused by HuR degradation. I specifically suggested they probe cell cycle markers and did show that M phase and S and G2 markers were indeed decreased in HuR trimmed cells, substantiating their claim that HuR knock down cells have impaired cell proliferation and therefore can be used as a tool to stall tumor growth in vivo. They did not perform the cell death assay I requested, however. While the apparently unchanged levels of housekeeping proteins such as GAPDH and Vinculin argued against it, it was by no mean a definitive proof. It is disappointing that the authors chose not to push the cellular toxicity a bit further but, as is, it should be sufficient for this publication. However, I advise the authors to look into that more substantially, especially if they push this system for clinical trials. The inadvertent delivery of HuR-BioPROTAC into healthy cells could cause serious issues (especially if Apoptosis is triggered) either as a results of HuR degradation directly or the degradation of some of its putative partners (ELAVL 2, IGF-BP3, TF-AP4) or putative HuR-bioPROTAC cross-reactants (ELAVL 2). Anticipating their root cause of these issues could prove valuable.

We thank the Reviewer for the constructive nature of their comments.

In one instance (item 14), the authors claimed to have introduced the requested change but did not. Supplemental figure 7 I (former supplemental figure 5I) was made using shades of yellow too pale to see. They are still the same color. I will leave it to the editor to make that decision.

Apologies for this oversight we have now changed the figure, as requested.

Overall, the authors have delivered an impressive amount of data on the development of a new tool to combat cancer. They made a fair attempt at answering my questions and concerns and the resulting manuscript is, in my opinion, fit enough for publication.

We thank the Reviewer for their thorough appraisal of our data.

Reviewer #2 (Remarks to the Author):

The authors have done well to address my comments. That mass spectrometry data analysis section is now clearer, and the additional information about various downregulated proteins in two out of three conditions is interesting and shows trends congruent with their original data. I can appreciate the challenges of generating a new doxycycline-inducible bioPROTAC cell line, and agree that applicability to other systems can be better addressed in follow up studies. With that, I see this manuscript as suitable for publication.

We thank the Reviewer for their thorough appraisal of our data.

Reviewer #3 (Remarks to the Author):

The authors have addressed my concerns.

We thank the Reviewer for their thorough appraisal of our data.

Reviewer #4 (Remarks to the Author):

The authors have addressed all my previous concerns.

We thank the Reviewer for their thorough appraisal of our data.